


**Transferability of machine learning-based modeling frameworks across flood events**
**for hindcasting maximum river flood depths in coastal watersheds**
Maryam Pakdehi[1,2], Ebrahim Ahmadisharaf[1,2*], Behzad Nazari[3], Eunsaem Cho[1,2]
[1]Department of Civil and Environmental Engineering, FAMU-FSU College of Engineering,
Tallahassee, FL 32310
[2]Resilient Infrastructure and Disaster Response Center, FAMU-FSU College of Engineering,
Tallahassee, FL 32310
[3]Department of Civil Engineering, University of Texas at Arlington, Arlington, TX 76010
**\*Corresponding Author**:
Dr. Ebrahim Ahmadisharaf
Research Faculty I
Department of Civil and Environmental Engineering
Resilient Infrastructure and Disaster Response Center
FAMU-FSU College of Engineering
Tallahassee, FL 32310, USA
Tel: +1 716-803-5498
Emails: eahmadisharaf@eng.famu.fsu.edu and eascesharif@gmail.com



**Abstract**
Despite applications of machine learning (ML) models for predicting floods, their transferability
for out-of-sample data has not been explored. This paper developed an ML-based model for
hindcasting maximum flood depths during major events in coastal watersheds and evaluated its
transferability across other events (out-of-sample). The model considered spatial distribution of
influential factors, which explain underlying physical processes, to hindcast maximum river flood
depths. Our model evaluation in a HUC6 watershed in Northeastern US showed that the model
satisfactorily hindcasted maximum flood depths at 116 stream gauges during a major flood event,
Hurricane Ida ($R^2$ of 0.92). The pre-trained, validated model was successfully transferred to three
other major flood events, Hurricanes Isaias, Sandy, and Irene ($R^2 > 0.71$). Our results showed that
ML-based models can be transferable for hindcasting maximum river flood depths across events
when informed by the spatial distribution of pertinent features and underlying physical processes
in coastal watersheds.
**Keywords**
Flood hindcasting; Machine learning algorithms; Maximum flood depth; Model transferability;
Coastal watersheds.
**1. Introduction**
Floods can damage civil infrastructure, business disruptions, and environmental degradation.
Nonstationary factors, including land use changes, population growth, and global warming, can
exacerbate the risk of flood events (Davenport, Burke, and Diffenbaugh 2021; National Academies
of Sciences, Engineering, and Medicine 2019; Galloway et al. 2018). For instance, (Galloway et
al. 2018) projected that changes in climate cause a 26.4% increase in the United States flood risks
by 2050. This increase in flood risk is expected to disproportionately affect poor communities,





leading to job losses and displacement of residents (Hino and Nance 2021). Therefore, effective
adaptation and mitigation strategies are urgently needed to maintain resilience against extreme
future floods (Hemmati et al. 2020; Qi et al. 2021; Wing et al. 2022).
To propose effective protection strategies, predictive models are used to evaluate watershed
responses under various plausible flood scenarios (Fernández-Pato et al. 2016; Kundzewicz et al.
2019; Viglione et al. 2014). These models are essential tools to inform decision makers about
suitable risk management strategies and actions. Flood models can be broadly categorized as
physically-based, morphologic-based and data-driven.
Physically-based models, widely used for predicting hydrologic events, are considered
reliable tools for assessing different flood scenarios (Fernández-Pato et al. 2016). These models
solve the shallow water equations to derive flood characteristics. Developing physically-based
models require certain meteorologic, hydrologic, and geomorphologic data. If these data are not
available at the desired scale, such models cannot be developed. For instance, global inundation
models are available across the globe, but they may not be efficient for small scale applications.
In such instances, data-driven models can be a flexible alternative as they can adapt to varying
levels of data availability by focusing on the features with sufficient data. This flexibility remains
one of the advantages of data-driven models over strictly data-dependent physically-based models.
Physically-based models also need significant computational resources, especially in the case of
high-resolution, multidimensional (2D and 3D) or stochastic models that necessitate numerous
simulations. To enhance the speed of flood simulations, techniques such as parallel computing,
graphics processing units (GPUs), and simplified models have been utilized (Costabile, Costanzo,
and Macchione 2017; Kalyanapu et al. 2011; Ming et al. 2020; Sridhar, Ali, and Sample 2021;


Zahura et al. 2020). However, resources for utilizing these approaches are not always available
(Zhang et al. 2014).

Morphologic-based models, which approximate flat-water surfaces over small spatial scales,

are also used for flood predictions (Bates 2022). Bathtub (Anderson et al. 2018; Kulp & Strauss
2019) and height above nearest drainage (HAND; Rennó et al. 2008) are two widely used models
in this modeling category. Jafarzadegan and Merwade (2019) used a probabilistic function based
on HAND, computed from a digital elevation model (DEM), and optimized it for accuracy, to
delineate 100-year floodplains. (Zheng et al. 2018) developed a synthetic rating curve using the
HAND method, which accurately represents the river shape and water level measurements, like
hydraulic models or stream gauge readings. While these models are computationally efficient, they
can overestimate flooded area and are limited to the number of features they use; these models rely
on topographic data (Bates 2022; Bates et al. 2005) and tend only to work well in confined valleys.
The sole use of topographic data makes HAND-based models impractical for low-lying areas,
especially coastal watersheds that experience a combination of hydrologic and oceanic processes
(tidal influences, storm surges and wave action); other flood influencing factors, which represent
such overlooked underlying physical processes, are needed along fore predictions in such
watersheds. Coastal regions experience a combination of oceanic and hydrological processes,
which might not be fully represented by HAND. Additionally, both HAND-based and bathtub
models are limited in representing such terrains as they might not fully capture the intricate
interactions between oceanic and hydrologic factors in coastal areas. Consequently, in coastal
watersheds, where unconfined floodplains and complex interactions are prevalent, alternative
modeling approaches that consider a broader range of factors are crucial for producing reliable



flood predictions. Incorporating these overlooked underlying physical processes becomes essential
in providing comprehensive flood predictions in these intricate environments.

Machine learning (ML) and deep learning (DL) models offer an alternative approach that can

rapidly capture complex relationships between various influencing factors and flood
characteristics. ML models have the potential to provide satisfactory predictions, making them a
valuable tool for improving flood prediction accuracy (Mishra et al. 2022). Such data-driven
models have gained popularity in overcoming the limitations of physically-based and
morphologic-based models in flood analyses (Khosravi et al. 2018). These models mathematically
represent the nonlinearity of flood dynamics using pertinent features and observed flood data, and
through their intricate nonlinear structures and algorithms. Data-driven models have been found
as promising tools due to their quick development time and minimal input requirements (Guo et
al. 2021; Löwe et al. 2021; Zahura et al. 2020); therefore, they are effective for short-term forecasts
and nowcasts (Mosavi, Ozturk, and Chau 2018). ML and DL models can discover and leverage
hidden patterns within the data, leading to improved performance as the amount of available data
increases. By recognizing and utilizing these underlying patterns inherent in the data, ML and DL
models can make satisfactory predictions (in terms of minimum error in estimating flood
characteristics like depth) and generate valuable insights. Example data-driven models for flood
prediction include multi-criteria decision-making techniques, multiple linear regression, artificial
neural networks (ANNs), random forest, convolutional neural networks, support vector machine,
support vector regression, frequency ratio models, and weights-of-evidence models (Adamowski
et al. 2011; Kim et al. 2016; Rafiei-Sardooi et al. 2021; Rahmati et al. 2016; Rezaie et al. 2022;
Wang et al. 2015; Youssef et al. 2022).



Previous research has shown that various ML algorithms are effective in predicting flood
extents and generating susceptibility maps, with a focus on classification ML models (Khosravi et
al. 2018; Rahmati et al. 2016; Rezaie et al. 2022; Youssef et al. 2022). However, these studies may
have limitations in terms of their experimental design and scope. For instance, some of these
studies created simplified datasets of flooded and unflooded points using remote sensing. The
datasets were often split into training and validation data, and different ML models were examined
on the same dataset. Another limitation of these ML studies is the reliance on a single event for
training and validation. These limitations call for studies that evaluate more complex
methodologies and a broader range of scenarios on the effectiveness of ML algorithms for
predicting flood characteristics.
Another application of ML models for flood inundation prediction has been incorporating them
with physically-based models for improving their performance. Such applications are based on the
hybrid use of ML and physically-based modeling categories. For instance, Chang et al. (2022)
suggested an approach that incorporated principal component analysis, self-organizing maps, and
nonlinear autoregressive models with exogenous inputs to mine spatiotemporal data and forecast
regional flood inundation. They recognized the value of using ML algorithms in conjunction with
a 2D hydraulic model to simulate urban flood inundation while taking different rainfall
occurrences into account. Elkhrachy (2022) developed a hybrid approach to predict flash flood
depths combining 2D hydraulic modeling with ML; water depths simulated by the Hydrologic
Engineering Center's River Analysis System (HEC-RAS; Brunner 2016) model served as inputs
to ML algorithms. Löwe et al. (2021) trained an ANN model to identify patterns in rainfall
hyetographs and topographic data to enable fast predictions of flood depths for new rain events
and locations (out of training sample data) complemented by 2D hydrodynamic simulations. Guo


et al. (2021) used a convolutional neural network model trained on flood simulation patch data
from the CADDIES cellular-automata model to perform image-to-image translation for rapid
urban flood prediction and risk assessment. To effectively simulate maximum flood extent and
depth, Hosseiny et al. (2020) created a system that combines a hydraulic model with ML
algorithms. Zahura et al. (2020) used simulations from high-resolution 1D/2D physically-based
models as training and test data for a random forest model that included topographic and
environmental characteristics to estimate hourly water depths. In these applications, flood depth,
which is important for risk assessments and damage estimates (Merz et al. 2010), has been
predicted by coupling physically-based and ML models. These coupled modeling studies
demonstrated the complimentary benefits of physically-based models along with ML algorithms
in producing flood modeling outputs, but the computational expense is still an application barrier.
Another significant challenge inherent in these studies lies in their dependence on 2D models for
training purposes. Furthermore, there appears to be a gap in demonstrating the ability of these
studies to successfully predict outcomes beyond their training samples. For instance, we are
unaware of studies that convincingly exhibit the capability of ML models to predict events of
greater magnitude than those utilized in their training datasets.

Despite previous efforts, the development of computationally efficient and user-friendly flood

prediction models remains a challenge. ML-based models, although promising and
computationally efficient, have not gained widespread acceptance among practitioners due to
concerns about their reliance on predicting flood characteristics for other events (out-of-sample).
While some studies have demonstrated promising results in generating flood hazard maps by
applying models to new geographical areas not used for training (Bentivoglio et al. 2022; Kratzert
et al. 2019; Zhao et al. 2021), few studies have examined the transferability of coupled ML and


physically-based models for predicting flood depths by applying them to unseen data not used in
training (Guo et al. 2021; Löwe et al. 2021). It, therefore, remains unclear whether an ML-based
model, which is trained, validated, and tested against a historical event, performs satisfactorily in
predicting flood characteristics of other events in the same watershed. Floods originate from
various sources, especially in coastal areas, where flooding heavily relies on the unique
characteristics of storm events. High wind events tend to generate storm surges that move
upstream, while intense rainfall over upstream watersheds leads to fluvial flooding that moves
downstream towards the coast. Conversely, slow-moving storm systems can cause intense local
rainfall, resulting in overland runoff entering rivers along their paths rather than a concentrated
upstream inflow flood wave. Hence, it is crucial to avoid overfitting an ML model to a single
historical flood event, as it can lead to significant underperformance in handling other events.
A further limitation of past research is the sole focus on predicting greatest flood extents using
classification-based algorithms, while the performance of regression-based ML models for
predicting other important characteristics like flood depths has not been investigated. Additionally,
the importance of spatial distribution of input features has been overlooked in past ML-based flood
modeling. To hindcast a flood characteristic at a given location, the features have been
incorporated at that location, but flooding is generated through contributions by several other
factors that are relevant across the upstream contributing watershed (in inland systems) and/or
from the downstream coastline (in coastal systems). The investigation of these research gaps
highlighted above is crucial to improve our capability in reliably hindcasting maximum flood
depths using computationally efficient and easy-to-use modeling frameworks.
This paper aimed to fill these research gaps by examining the performance and transferability
of ML models in hindcasting maximum flood depths across various events in a coastal watershed.




Our objective was to develop a transferable, computationally efficient model to hindcast flood
depths. To achieve this, the study developed a modeling framework based on an ML algorithm.
The developed ML-based model combined the ANN algorithm with feature selection methods and
geospatial data. We evaluated the performance of this model against one extreme flood event,
Hurricane Ida, across a coastal watershed (HUC6)—Lower Hudson—in Northeastern US. Next,
we assessed the transferability of our developed model across three other extreme events—
Hurricanes Isaias, Sandy, and Irene—in the same watershed. These events encompass varied
rainfall intensities, wind speeds and storm track directions. Unlike past ML-based modeling
studies, which focused solely on predicting flood status (flooded or unflooded), our regression-
based model estimates maximum flood depths. This model was also examined against multiple
events, more than one single event that has been the focus of past research (Bafitlhile and Li 2019;
Dawson et al. 2006; Hosseini et al. 2020). The model also considered the spatial dimension for
predicting flood depths at a given location, in which the features were represented either at that
location or across the contributing watershed. This ML model is generic and can be applied to
hindcast flood depths at non-gauge river sites to get a denser reconstruction of an event along the
river network and hindcast water levels in watersheds with similar drainage area (HUC6 or larger)
and flood type (fluvial and coastal).
**2. Methodology**
We developed an ML-based model that hindcast maximum flood depths at stream gauges
across a coastal watershed during an event (Figure 1). A coastal watershed receives flood
contributions from the inland and coastal systems (i.e., fluvial, and tidal). The model uses
geospatial analyses and ML algorithms to hindcast maximum flood depths during an event at river
cross-sections of a given watershed. This model is informed by underlying physical flood




processes represented by a wide array of features (topographic, meteorologic, hydrologic, land
surface, soil and hydrodynamic).

Geospatial operations were conducted to compute the features at stream gauges and/or over

their contributing watersheds (the upstream area that drains water to the gauge) with a careful
consideration of underlying physical processes. We used feature selection techniques to determine
the most key features and used those in our ML model. Applying observed data from stream gauges
during a flood event, the model was trained, cross-validated and tested. We then evaluated the
model transferability by examining its performance in three other extreme flood events.

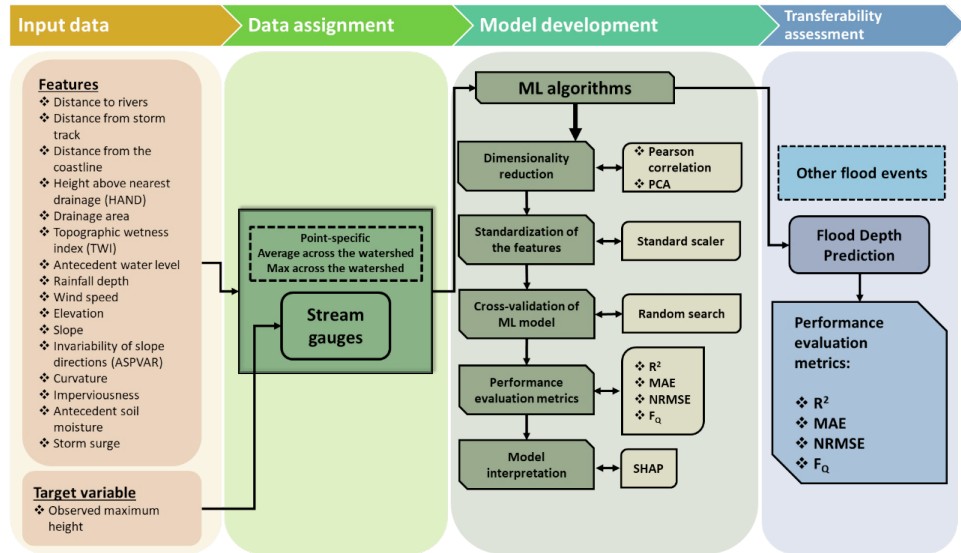


Figure 1: Schematic view of the machine learning (ML)-based model for hindcasting maximum
flood depths in coastal watersheds. ANN: Artificial neural network; PCA: Principal component

analysis; SHAP: Shapley additive explanations; MAE: Mean absolute error; NRMSE:

Normalized root mean square error; $F_Q$: ratio of estimated over observed maximum flood depth.


**2.1. Selection and calculation of key features**
When developing an ML model, the features play a pivotal role in determining its performance
and estimation capability. By selecting the most relevant and representative features, we empower
the model to discern the underlying patterns and relationships within the data more accurately. The
ultimate objective is to enable the model to comprehend the complexities associated with flooding,
a phenomenon influenced by a myriad of interrelated factors. For an ML estimation accuracy to
be transferable for complex physical phenomena of flooding, the selection process should extend
beyond merely choosing features based on their individual statistical significance. Instead, it
should focus on identifying features that collectively contribute to a holistic representation of the
phenomenon. This approach ensures that the ML model can generalize well to unseen data and
handle various real-world scenarios effectively. By incorporating this comprehensive set of
features, the ML model can capture the nuanced interactions between these features; this enhances
the model estimation performance.
We selected key features for our ML-based flood model according to the existing research and
the underlying physical processes. Our model considers these features from five broad categories
of geographic location, hydrologic, topographic, land surface, soil, and hydrodynamic (Table 1).
Here, we provide information on how to derive the features to hindcast flood depths during a flood
event in a coastal watershed. Aside from the soil category that represents pre-flood conditions
(antecedent soil moisture), all other features represent conditions during a flood event.




Table 1. Machine learning model features and the assignment approaches for stream gauges.

| Category | Feature | Point-specific | Spatial average across the contributing watershed | Spatial maximum across the contributing watershed |
|---|---|---|---|---|
| **Geographic location** | Distance to rivers | | * | |
| | Distance from storm track | * | | |
| | Distance from the coastline | * | | |
| **Hydrologic** | Height above nearest drainage (HAND) | | * | |
| | Drainage area | * | | |
| | Flow accumulation | * | | |
| | Topographic wetness index (TWI) | * | * | |
| | Antecedent water level | * | | |
| **Meteorologic** | Rainfall depth | * | * | * |
| | Wind speed | * | * | * |
| **Topographic** | Elevation | * | | |
| | Ground slope | * | * | |
| | Slope aspect | * | * | |
| | Slope aspect invariability (ASPVAR) | | * | |
| | Curvature | * | * | |
| **Land surface** | Imperviousness | | * | |
| **Soil** | Antecedent soil moisture | * | * | |
| **Hydrodynamic** | Storm surge | * | * | |


By integrating all these factors into our methodology, we developed a flood hindcast model
that accounts for key processes in a coastal watershed. We used a two-step process to assign feature
values to a point located on a stream gauge. Depending on the feature, we assigned specified values
to the gauge itself or its contributing watershed to consider the spatial dimension in flood
generation processes. For the contributing watershed, spatial mean, and maximum across the
contributing watershed of a given stream gauge was computed. This method ensures that the
feature values indicate the overall pertinent physical processes occurring at the streams and
upstream watersheds. Table 1 specifies how each feature was used in our model.

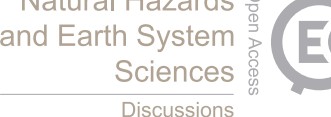

For features under the geographic location category, we incorporated distance to rivers—
critical factor in determining flood risk in numerous studies (Cao et al. 2020; Rafiei-Sardooi et al.
2021), storm track—specific to the flood event from (National Hurricane Center 2022)—and
distance to the nearest coastline. The proximity of a location to waterbodies, such as rivers or
coastlines, directly influences its vulnerability to flooding. Coastal regions are susceptible to storm
surges, which occur during tropical storms or hurricanes. Storm surges are massive walls of
seawater that get pushed ashore by intense winds. As a result, coastal areas can experience severe
flooding. Storm tracks, however, are pathways in the atmosphere along which storms, such as
hurricanes, tropical cyclones, or extratropical storms, tend to move. These storms often carry heavy
rainfall, intense winds, and storm surges, which can lead to severe flooding in areas they pass over
or affect. The distance to storm track and coastline is both considered "Point-specific" as they are
specific to individual locations. However, distance to rivers is identical (zero) at these stream
gauges, but different in the contributing watersheds, so we calculated the spatial average distance
of the contributing watersheds to the rivers.
Under the hydrologic category, we employed four variables of HAND, drainage area, flow
accumulation, topographic wetness index (TWI), and antecedent water level. HAND represents
the elevation of a location relative to the nearest stream. This feature is widely used in flood
modeling due to its ability to hindcast flood-prone areas by considering topography and water flow
characteristics (Hu and Demir 2021). As its value at the stream gauges is zero, its spatial average
across the contributing watershed was considered. The drainage area provides information about
potential runoff, while flow accumulation feature helps predict water flow paths during flood
events that is previously used by Löwe et al. (2021) and Pham et al. (2021). Both drainage area
and flow accumulation values at point of stream gauge (Point-specific) were captured. TWI was




calculated using Equation (1) based on the ground slope and drainage area of the contributing
watershed (Beven and Kirkby, 1979), and was used by (Gudiyangada Nachappa et al. 2020; Löwe
et al. 2021; Pham et al. 2021; Zahura et al. 2020; Zhao et al. 2020).
$$TWI = \ln\left(\frac{\alpha}{\tan(\beta)}\right)$$    (1)
where, $\alpha$ is the upslope contributing area per unit contour length (as known as the specific
catchment area), and $\beta$ is the local slope gradient in radians. Its value was considered for both
"Point-specific" and "spatial average across the contributing watershed" to represent the specific
location and the overall characteristics of the contributing watershed. The last feature in this
category is antecedent water level which refers to the gauge height one day before the event as was
considered "Point-specific" for stream gauges.
The meteorologic category features were precipitation (Rafiei-Sardooi et al. 2021) and wind
speed. Rainfall is the main driving force for floods (Mishra et al. 2022). Storms can bring intense
and prolonged rainfall to certain areas. If a storm passes over or near a location, it can result in
excessive precipitation, overwhelming local drainage systems and causing flooding in low-lying
or poorly drained areas. Wind speed is another key feature that can influence the severity and
extent of flooding, especially in the context of hurricanes. Intense winds during storms and
hurricanes generate large and powerful waves in the ocean. These waves can exacerbate the impact
of storm surges, causing even more coastal flooding as they crash onto the shore and flood areas
even farther inland. We obtained daily precipitation and wind speed data for the entire period of
flood event from weather stations of the National Oceanic and Atmospheric Administration
National Centers for Environmental Information (NOAA's NCEI 2022). Their maximum values
over a flood event were computed at each station. Using point-based precipitation and wind speed




data, we then created a spatially distributed rainfall and wind speed dataset by interpolating the
maximum values using the Inverse Distance Weighting (IDW) method (Hosseini et al. 2020).
Rainfall depth and wind speed are considered for " Point-specific," "spatial average across the
contributing watershed," and "spatial maximum across the contributing watershed." These values
capture the intensity of the meteorological conditions at individual points and the overall average
and maximum values across the watershed.

Elevation, ground slope, slope aspect, aspect invariability (ASPVAR), and curvature were

features under the topographic category (Cao et al. 2020; Chen et al. 2023; Huang et al. 2022;
Khosravi et al. 2018; Rafiei-Sardooi et al. 2021; Sun et al. 2020). DEM with a resolution of 1/3
arc-second (~10 m) was acquired from the United States Geological Survey (USGS 2022). To
remove any fake depressions, the DEM sinks were filled. Before beginning any hydrological study
with DEM data, this is a suggested step that is frequently employed (Khosravi et al. 2018; D. Zhu
et al. 2013). Elevation, ground slope, slope aspect, invariability of slope directions (ASPVAR),
and curvature all were derived from DEM. Elevation allows us to identify low-lying regions prone
to floods and hindcast the flood depths. Ground slope is one of the most key factors in water
movement. The slope of the land, also known as the topography or gradient, plays a crucial role in
determining the direction and velocity at which water flows across the landscape. On sloped
terrain, water flows along the path of least resistance, which is typically downhill. The angle of
the slope determines the speed and volume of surface runoff, influencing the potential for flooding.
Slope aspect provides insights into surface runoff distribution and flow concentration by indicating
the direction that each slope faces affects hydrologic processes (Gudiyangada Nachappa et al.
2020; Rafiei-Sardooi et al. 2021). Similar to (Gudiyangada Nachappa et al. 2020), we divided
slope aspect into 10 categories: north (0°-22.5°; 337.5°-360°), northeast (22.5°-67.5°), east (67.5°-





112.5°), southeast (112.5°-157.5°), south (157.5°-202.5°), southwest (202.5°-247.5°), west

(247.5°-292.5°), northwest (292.5°-337.5°), and flat (0°). ASPVAR values near zero indicate

diverse catchment slope aspects, while values approaching 1.0 imply a dominant direction (Wan

Jaafar and Han, 2012). This feature provided information about surface runoff distribution and

flow concentration by specifying the direction water would flow across the terrain (Dawson et al.

2006). Additionally, analyzing the curvature helped us understand how it impacts flood events, as

the topographic curvature plays a role in determining the flow of runoff (Khosravi et al. 2018;

Pradhan 2009). Elevation is considered "Point-specific", while ground slope, and curvature are

considered for both "Point-specific" and "spatial average" across the contributing watershed,"

indicating how these topographic features vary throughout the entire watershed. ASPVAR

conceptually represents the "spatial average across the contributing watershed," capturing the

overall characteristics of watersheds.

The land surface category was represented by only one variable, imperviousness. On

impervious surfaces, that reduce the ability of soil to absorb rainfall via infiltration, larger volumes

of surface runoff are produced and propagated downstream. In fact, impervious surfaces increase

both the quantity and velocity of runoff, and this is due to their higher surface smoothness and

lower friction to resist water movement. This rapid flow of water can overwhelm natural

waterways, increasing the risk of flooding. We used the spatial average of imperviousness across

the contributing watershed in the model.

Soil category included antecedent soil moisture, which reflects the pre-storm saturation extent,

essential for runoff estimates and high moisture flux production from rain-bearing systems

(Ahmadisharaf et al. 2016; Jafarzadegan et al. 2023; Mishra et al. 2022). It is calculated over one

day before the storm and considered for both "Point-specific" and "spatial average across the





contributing watershed." These values indicate the stream gauge surrounding content and its
average value over the entire watershed.

In the hydrodynamic category, we used storm surge from tidal gauges on the coast. Storm

surge was estimated as the difference between the maximum water level and the astronomical tide
during a flood event that was downloaded from NOAA ("NOAA Tides & Currents" 2023). This
feature is crucial in hindcasting the impact of coastal contributions to flood events. If the flood
event does not receive any coastal contributions, this category can be removed from the list of
model features. It is considered for both "Point-specific" and "spatial average across the
contributing watershed" presenting the stream gauge and its entire watershed tidal condition.

2.1.1 Feature selection method

We employed common feature selection methods, such as Pearson's correlation coefficients

(Cao et al., 2020; Chen et al., 2023; Lee et al., 2020) and principal component analysis (PCA) – a
widely used technique in many studies (Abdrabo et al., 2023; Chang et al., 2022; Reckien, 2018)
to identify most important features for hindcasting flood depths of a given event in a watershed.
The PCA components were evaluated based on their absolute values, allowing us to quantify the
contribution of each feature to the overall variance. By summing the absolute values across all
features, we obtained importance scores for each feature, which enabled us to rank them in
descending order. While the Pearson's correlation coefficients are tailored for assessing linear
relationships, the PCA captures both linear and non-linear relationships. The strength and direction
of linear relationships between the features and flood depth were evaluated using Pearson's
correlation coefficient. Through PCA, we determined which principal components in the feature



set captured the most variation. These analyses enabled us to narrow down the initial list of the
features.

**2.2. Machine learning (ML) models**
2.2.1. Artificial neural networks (ANNs)

To hindcast the flood depth, the target variable, we employed ANN. This algorithm was trained

via observed flood depths from stream gauges using the key features selected through our feature
selection (Section 2.1). The choice of ANN was based on previous successful applications in
complex environmental modeling problems (e.g., Adedeji et al., 2022), including flood depth
estimations (e.g., Dawson et al., 2006) (Abrahart, Kneale, and See 2004; Bafitlhile and Li 2019;
Berkhahn, Fuchs, and Neuweiler 2019; Dawson et al. 2006; Rumelhart, McClelland, and Group
1986; J.-J. Zhu, Yang, and Ren 2023). One of the key advantages of using ANN is its capacity for
generalization, as highlighted by Maier et al. (2023), allowing the model to perform well on unseen
data, making it robust and reliable for real-world flood estimations. Additionally, ANN has been
used in flood estimations due to its ability to determine the relationship between rainfall and runoff
without relying on specific physical processes, thus addressing the complexities and limitations
encountered in hydrologic models (Bafitlhile and Li, 2019). ANNs are computing systems inspired
by the biological neural networks that constitute animal brains (Dawson et al., 2006, p. 200;
McCulloch and Pitts, 1943). They are designed to simulate the behavior of biological systems
composed of "neurons". ANNs are composed of nodes, or "artificial neurons", connected and
operate in parallel. Each connection is assigned a weight that represents its relative importance.
During the learning phase, the network learns by adjusting these weights based on the input data
it is processing (McCulloch and Pitts, 1943). ANNs have also been widely utilized in flood





estimations due to their ability to model complex relationships and their tolerance for noisy data.
Considering the robustness, accuracy, and proven success of ANN in flood estimation tasks, it was
deemed suitable for our flood depth estimations. Here, ANN was implemented using python's
Keras library with TensorFlow backend.

2.2.2. Machine learning (ML) model pre-processing and implementation
The observed flood data and features were split into training and testing sets, with 70% to 90%
of the data used for training and 10% to 30% for testing (Joseph 2022; Nguyen et al. 2021). The
numerical features in the data were standardized using the StandardScaler function from the Scikit-
learn library of python. Hyperparameter optimization is a step in improving the performance of
ML models. This process involves identifying the optimal hyper-parameter values for ML
classifiers. We used the Random Search cross-validation approach (Boulouard et al. 2022; Hashmi
2020) to perform hyper-parameter optimization. This approach performs a randomized search on
hyperparameters using cross-validation. The hyperparameters we optimized here included the
number of layers, units, activation functions, optimizer, regularization rate, batch size, and epochs.
The best hyperparameters were selected based on the negative mean squared error. The ANN
model was trained using the training data and the best hyperparameters obtained from the
optimization process. To prevent overfitting, we used early stopping and model checkpointing
during the model training. Early stopping was implemented to stop training when the validation
loss stopped improving, and model checkpointing was used to save the model with the lowest
validation loss. Cross-validation was performed using a 5-fold cross-validation strategy during the
hyperparameter optimization process. This strategy involved splitting the training data into five
subsets and training the model five times, each time using a different subset as the validation set.



We allocated 90% of the data for training and 10% for testing. While the portion for test is small,
the utilization of cross-validation, randomized hyperparameter search, early stopping, and model
checkpointing collectively works to construct a model less susceptible to overfitting on a particular
test set. This allocation of 10% for testing, combined with these methodologies, is designed to
enhance the model's ability to generalize across diverse scenarios.

2.2.3. Model performance evaluation
The performance of the ANN model was evaluated using coefficient of determination ($R^2$),
Mean Absolute Error (MAE), Normalized Root Mean Square Error (NRMSE), and the ratio of
estimated over the observed maximum flood depth ($F_Q$; Schubert and Sanders 2012). The R2
metric measures the proportion of variance in the dependent variable predictable from the
independent variables. The MAE measures the average magnitude of the errors in a set of
estimations without considering their direction (i.e., overestimation or underestimation). The
NRMSE is a metric that quantifies the normalized average magnitude of the prediction error. It
assesses the relative size of the root mean square error (RMSE) by considering the RMSE in
relation to the average of the observation. It is commonly used in regression analysis and a smaller
NRMSE value indicates a higher level of agreement between the estimated values and the actual
observations (Stow et al. 2003; Ahmadisharaf Ebrahim et al. 2019). These metrics were calculated
for both training and testing datasets to assess the model performance.



### 2.2.4. Model interpretation

To interpret the model and understand the contribution of each feature to the estimation, we used SHapley Additive exPlanations (SHAP) that is a game theoretic approach to explain the output of an ML model (Lundberg and Lee, 2017). It connects optimal credit allocation with local explanations using the classic Shapley values from game theory and their related extensions. The SHAP values interpret the impact of having a certain value for a given feature in comparison with the estimations we would make if that feature took some baseline value (Abdollahi and Pradhan, 2021). In other words, SHAP estimates how much each feature contributes to the predictive model output for a particular instance. The SHAP results on the feature importance and their impacts on the model estimation can be presented using a plot to visually show the distribution of impacts of each feature on the model output. A positive SHAP value indicates that the feature's presence increases the model output, while a negative SHAP value indicates that it decreases the model output.

**2.3. Model transferability across flood events**

The ML-based model, which was initially developed, trained, and validated based on one flood event, was subsequently examined as is (with no additional parameter tuning) against other events in terms of the performance and generalizability in hindcasting maximum flood depths. By examining our model against different flood events, we aimed to evaluate its effectiveness in hindcasting flood depths across diverse events. This evaluation allowed us to assess the ML model's ability to handle varying flood conditions and its potential for application in different events in the same watershed.



### 3. Study area

The study area is the Lower Hudson Watershed a six-digit hydrologic unit code (HUC 020301) according to the USGS classification. The 10,068 km$^2$ watershed is in the Northeastern United States (Figure 2) spanning parts of three states, Connecticut, New Jersey, and New York. This watershed has a humid subtropical climate with hot summers and mild winters. The highest elevation is ~450 m above mean sea level. Residential, agriculture, and forest are the dominant land uses in the watershed according to the 2022 National Land Cover Dataset (NLCD) (USGS 2022). Large metropolitan areas like New York are in the study watershed. The population density was estimated at 344 persons per square km in 2020 (USCB, 2020), with higher concentrations in urban areas like New York and lower densities in rural parts. Several major rivers drain into the watershed, including the Hudson River, which flows for 496 km (about the length of New York State). The ground slope varies from 87.5% in the mountainous parts to 0% in the coastal region.

We studied four major flood events in the study area. The primary event for model development was Hurricane Ida in 2021, while three other hurricanes—Isaias (2020), Sandy (2012) and Irene (2011)—were used to assess the model transferability. Hurricane Ida, a devastating Atlantic Category 4 hurricane that made landfall in September 2021, hit Louisiana, and progressed toward the Northeastern United States. The hurricane caused considerable floods and significantly impacted both the west-south-central region, including New Orleans, and the northeastern region, with severe damages reported in New York City and Philadelphia (Beven II, Hagen, and Berg 2022; J. Wang et al. 2022). The storm remnants sent record-breaking rainfall to the New York region as they headed northeast, resulting in flash flooding (Beven II, Hagen, and Berg 2022). The extensive flooding and severe property destruction caused by Hurricane Ida's record-breaking rains highlighted the importance of comprehending the hurricane effects on



affected areas. Furthermore, strengthening regional resilience to catastrophic flooding episodes
requires the development of effective mitigation strategies. The three other events, which were
used to evaluate the model transferability, were also most recent major hurricanes after 2000 with
available stream gauge data and differing track and intensity. In 2020, Hurricane Isaias, a Category
1 hurricane, made a quick trip along the East Coast, bringing with it severe rain and floods,
especially in the Mid-Atlantic and Northeast. The storm's rapid passage caused several deaths and
extensive power losses (Latto, Hagen, and Berg 2021). In 2012, superstorm Sandy, commonly
known as Hurricane Sandy, struck the Northeast and caused severe damage. It produced significant
flooding due to the intense storm surge and torrential rains, especially in New York and New
Jersey, where the storm surge reached record heights (Blake et al. 2013). In 2011, a huge and
catastrophic storm named Hurricane Irene affected a major portion of the Eastern Seaboard. Heavy
rains from the storm caused significant flooding, especially in Vermont, where it was the worst
flooding in over a century for that state (Lixion A. and Cangialosi 2013).

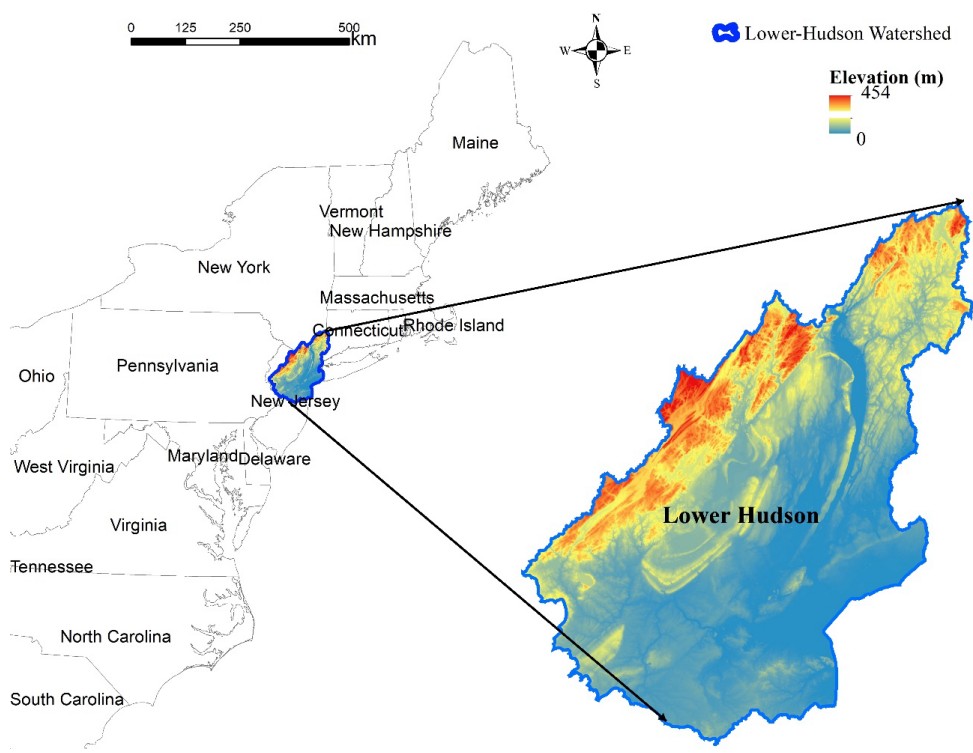

Figure 2. Lower Hudson River Watershed.

**3.1. Data collection**

Table 2 lists the data used for the study area alongside their source and spatial and temporal resolutions. We acquired instantaneous stream gauge height data from the USGS's National Water Information System to analyze water levels during the four flood events. While the features' data had different spatial resolutions, we did not make them consistent because only at-point (stream gauges) or aggregated spatial statistics of contributing watersheds were used in the ML model; no combinations of the features were needed.



The study watershed embraces 116 stream gauges, seven weather stations and two tidal gauges
(Figure 3). These gauges and stations recorded the data for all the four events (Hurricanes Ida,
Isaias, Sandy, and Irene). The drainage area of the contributing watersheds of the stream gauges
varies from 5.5 to 2,104 km². The range of maximum recorded flood depths, rainfall, and
antecedent soil moisture at the stream gauges during the four hurricanes are presented in Table 2.
It shows that Hurricane Ida had a narrower range of water levels, even though it generated lower
cumulative rainfall depths. In contrast, Hurricane Irene had the broadest range in river water levels,
likely due to the significant amount of rainfall it encountered during the event. Also, Ida and Irene
had similar antecedent soil moisture conditions, which could have influenced their respective river
water levels. Hurricane Sandy had a higher antecedent soil moisture percentage range of 17% to
38% compared to both Ida and Isaias, indicating a potentially higher level of saturation before the
storm's arrival. This may have contributed to Sandy's significant storm surge, which ranged from
1.97 to 2.85 m, compared to Ida and Isaias with storm surge ranges of 0.25 to 0.67 m and 0.20 to
0.76 m, respectively.

Table 2. The range of river water level, cumulative rainfall depth and antecedent soil
moisture in the flood events.

| Hurricane | Year | River water level (m) | Cumulative rainfall depth (mm) | Antecedent soil moisture (%) | Storm Surge (m) | Wind Max (m/s) | Distance to storm track (m) |
|---|---|---|---|---|---|---|---|
| Ida | 2021 | 0.85-36.66 | 0.01-45.43 | 21-43% | 0.25-0.67 | 27.64-35.49 | 0.09-1.1 |
| Isaias | 2020 | 0.22-35.35 | 17.37-62.22 | 9-39% | 0.20-0.76 | 48.29-65.33 | 0.23-1.14 |
| Sandy | 2012 | 0.24-35.98 | 19.83-56.53 | 17-38% | 1.97-2.85 | 63.43-76.97 | 0.77-2.16 |

Natural Hazards and Earth System Sciences
Author(s) 2023





| Irene | 2011 | 1.03-37.33 | 147.29-217.74 | 19-43% | 1.05-1.37 | 51.05-60.68 | 0.00-0.93 |


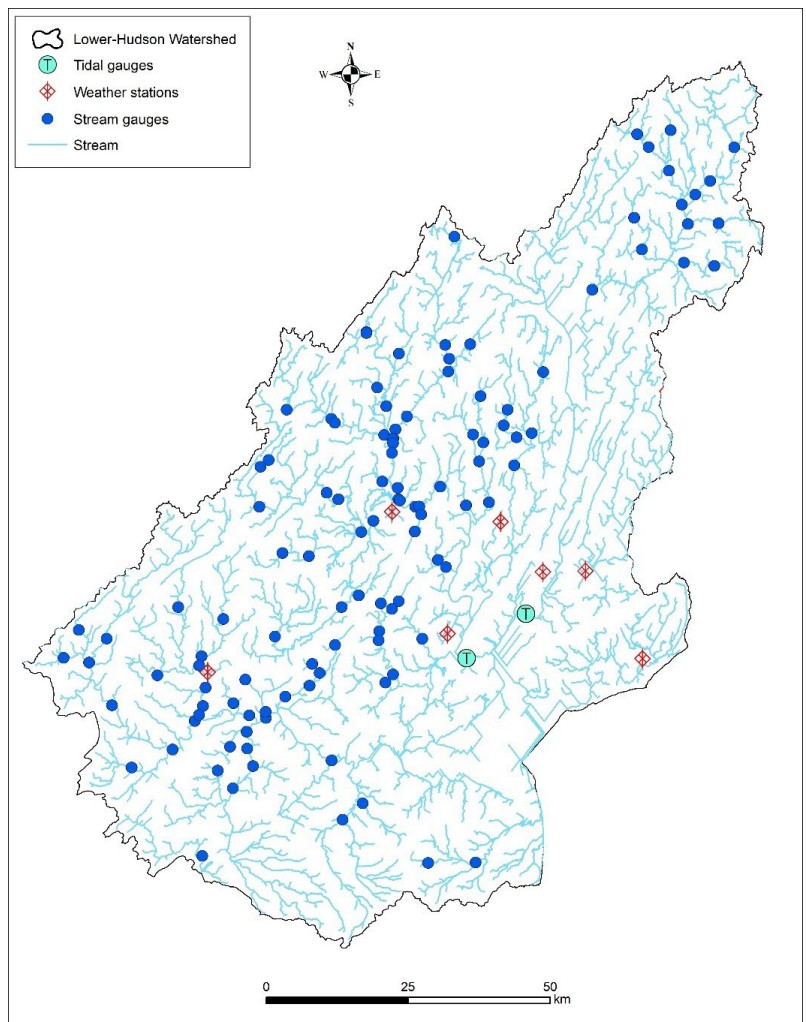


Figure 3. Stream and tidal gauges and weather stations in the study watershed.



Table 3: Model features and data sources and resolutions in the study area. NHDPlus -
National Hydrography Dataset Plus; NED - National Elevation Dataset; USGS NWIS - United



States Geological Survey National Water Information System; NCEI - National Centers for
Environmental Information; NLCD - National Land Cover Database; ERA5 - Fifth Generation of
the European Centre for Medium-Range Weather Forecasts (ECMWF) Reanalysis; NOAA -
National Oceanic and Atmospheric Administration.

| Category | Feature | Source | Spatial resolution | Temporal resolution |
|---|---|---|---|---|
| **Geographic location** | Distance to rivers | | — | — |
| | Distance from storm track | NHDPlus | — | — |
| | Distance from the coastline | | — | — |
| **Hydrologic** | Height above nearest drainage (HAND) | NED | 10 m | — |
| | Drainage area | | — | — |
| | Flow accumulation | | — | — |
| | Topographic wetness index (TWI) | | — | — |
| **Meteorologic** | Rainfall depth | NCEI | — | Daily |
| | Wind speed | | | |
| **Topographic** | Elevation | NLCD | 10 m | — |
| | Ground slope | | | — |
| | Slope aspect invariability (ASPVAR) | | | — |
| | Curvature | | | — |
| **Land surface** | Imperviousness | NLCD | 30 m | — |
| **Soil** | Antecedent soil moisture | ERA5 | — | Daily |
| **Hydrodynamic** | Storm surge | NOAA Tides and Currents | — | Sub-hourly |


Figure 4 displays the variations in water levels and storm tracks for all hurricanes. The total
slope aspect is south, which results in shallower depths at the upper point of the river. As we
move southward along the river's mainstream, deeper water levels are observed.




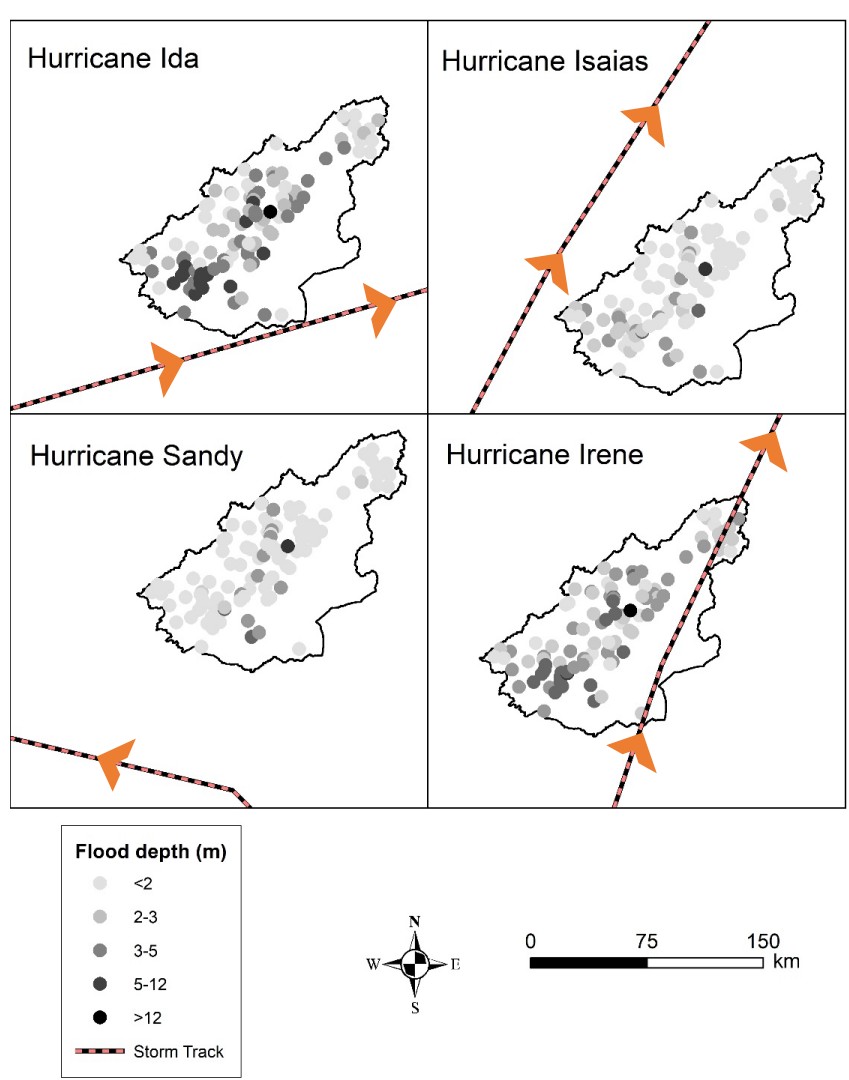


Figure 4. Water levels across the study area during studied hurricanes.



## 4. Results and discussion

### 4.1. Feature selection

4.1.1. Pearson's correlation matrix

As a result of Pearson's correlation analyses, we eliminated five features with absolute correlation coefficients greater than 0.70, the cutoff threshold suggested in previous studies (Cao et al. 2020; Chen et al. 2023; Lee et al. 2020). The strong correlation coefficient of 0.99 between "Drainage area" and "Flow accumulation" indicated that both variables capture similar information about water flow and storage in the watershed. To avoid collinearity issues, "Flow accumulation" was excluded from further analyses. Similarly, the high correlation coefficient of 0.97 between "Rain-MAX" and "Rain-Mean" suggested that they offer similar information about maximum and average rainfall values across the watershed. Consequently, "Rain-Mean" was excluded from consideration. Additionally, a correlation coefficient of 0.94 between "Tide-Mean" and "Tide-Point" indicated that the average tide level within the watershed closely resembled tide levels measured at stream gauge points. As a result, "Tide-Point" was excluded from the analysis. By considering the correlation coefficients and the potential redundancy among features, we ensured that independent variables, which are essential for modeling flood depths, are selected.

4.1.2. Principal Component Analysis (PCA)

We conducted PCA to assess the importance of various features in hindcasting flood depths. The results of the PCA analysis unveiled the key features that significantly influence the flood depth.

Interestingly, we identified the "Slope-Point", river slope at the stream gauges, "Slope-Aspect," and distance from the coastline as the least key features for capturing the overall variability



of maximum flood depth. Consequently, we excluded it from further analyses. The lesser
importance of "Slope-Point" and "Slope-Aspect" may be since river slope is related to bathymetry,
which is typically not represented well by DEMs (Bhuyian and Kalyanapu 2020).

**551     4.2. Machine learning (ML) model development**

4.2.1. Model development and performance evaluation
We conducted a thorough hyperparameter optimization process to fine-tune the neural network
model for estimating the flood depth of Hurricane Ida. The optimization process involved 500 fits,
with each fit considering 100 candidates for each of the five folds in the cross-validation. This
helps to ensure that the model's performance is robust and not dependent on a specific
training/testing split. As a result, the model became more effective in making estimations on
unseen data, as indicated by the enhanced testing performance. Furthermore, the optimization
process allowed us to find the best combination of hyperparameters that optimized the model's
performance. The best hyperparameters were identified as follows: 50 units, a regularization rate
of approximately 0.104, the sgd optimizer, one layer, 600 epochs, a batch size of 8, and the elu
activation function. These optimized hyperparameters were then used to train the ANN model and
evaluate its performance. This meticulous hyperparameter optimization approach ensured that the
model was fine-tuned to achieve the best possible performance for estimating flood depths.
The model demonstrated excellent performance on the training dataset, with an $R^2$ of 0.93,
indicating that the model can explain 93% of the variance in the training data. The MAE for the
training data was 0.64 m, and NRMSE was 28%, suggesting that the model estimations were
satisfactory. On the test dataset, the model achieved an $R^2$ of 0.87, MAE of 0.87 m, and the
NRMSE was 33%. These values also show that the model's performance was satisfactory during




the test phase but slightly poorer than the train phase. The training history plot showed that the
model performance improved with each epoch during training, indicating that the model was
learning from the data. The model training process stopped at epoch 75 due to early stopping.

4.2.2. Model interpretation
Figure 5 provides an overview of the influence of distinctive features on the model estimation
on flood depths. The SHAP values measure the contribution of a feature to the estimation for each
sample in comparison to the estimation made by a model trained without that feature.

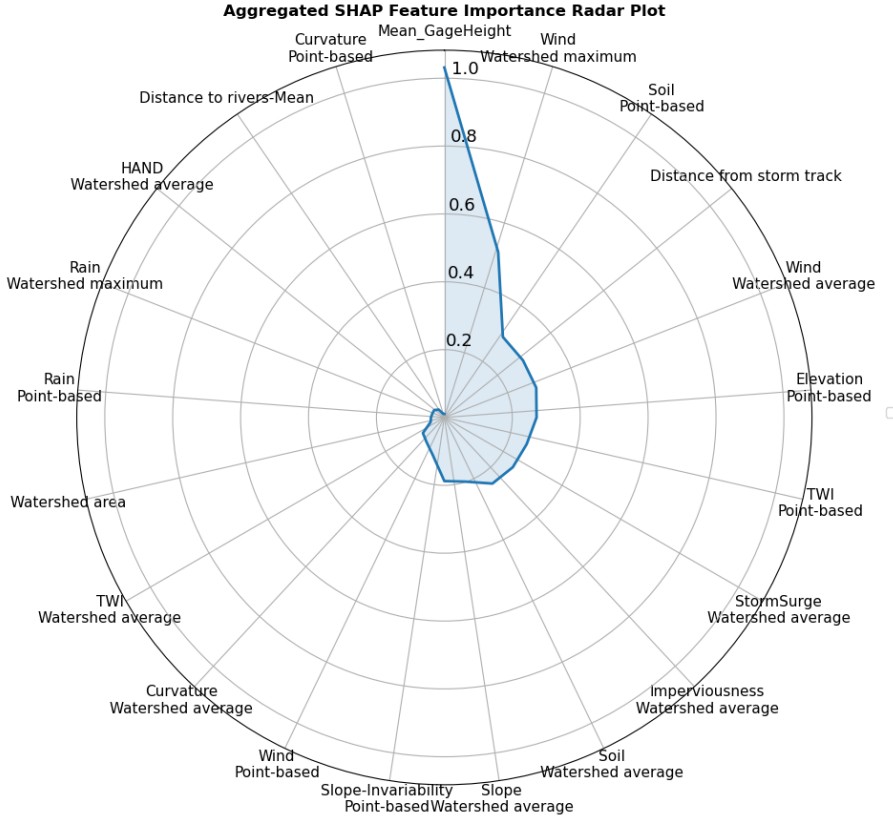


Figure 5. Shapely additive explanations (SHAP) summary plot of the flood model.

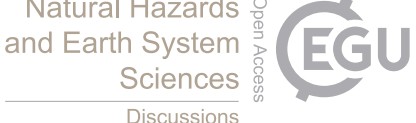


The most influential features in estimating flood depths are antecedent water level, indicating that streams with higher water levels before an event are subject to greater flood depths. When combined with additional rainfall or water input during a flood, they lead to increased flood depths. Similarly, spatial maximum wind speed across the contributing watershed, antecedent soil moisture at point, and elevation are other significant factors affecting flood depth estimations, with greater values associated with higher estimated flood depths. Intense winds during a hurricane accelerate the movement of floodwaters, leading to greater depths in certain areas, while saturated soil has limited capacity to absorb additional water, resulting in more surface runoff and higher flood depths. The inclusion of elevation as an important feature in our study closely aligns with the findings of Hosseini et al. (2020) and Chen et al. (2023) in their flash flood susceptibility and hazard assessment one on a small non-coastal watershed and the other on a large coastal watershed. Elevation has been consistently recognized as a crucial factor influencing flood occurrences, as it directly affects the water flow and drainage patterns within a watershed (Rafiei-Sardooi et al. 2021).

On the other hand, features such as the spatial average of distance to rivers across the contributing watershed, the spatial average of HAND across the contributing watershed, and rainfall both at point and the spatial maximum of it across the watershed were identified as the least key features in estimating flood depths. This can be attributed to the fact that our target is hindcasting flood depths at stream gauges, while these input features are more associated with flood depths occurring away from the stream network. Consequently, these features exhibit a limited impact on the model predictive performance when compared to other factors. The spatial average of distance to rivers and HAND have limited variability within our watershed and might





not fully capture relevant information about geography, topography, and drainage patterns, leading
to reduced discriminatory importance in flood depth estimation models.
The finding about the less importance of rainfall in flood estimation concurs with the results
reported in the study by Salvati et al. (2023) in pinpointing vulnerable regions within a non-coastal
medium-sized watershed. The study suggests that rainfall may have a lower impact on flood
occurrences or flood depth estimations compared to other influential factors. This highlights the
significance of considering a comprehensive set of variables in flood modeling to accurately
capture the underlying relationships and improve estimation performance. The model ability to
capture these complex relationships demonstrated its potential utility in flood estimation and
management.

**4.3. Examining the machine learning (ML) model transferability across flood events**

The transferability of the trained and tested model (against Hurricane Ida) was examined by
applying it to three other events within the same watershed. Table 4 summarizes the evaluation
metrics for the three hurricanes.

Table 4. Model performance across in historical flood events. MAE - mean absolute error;
RMSE - root mean square error, $F_Q$ - ratio of estimated over observed maximum flood depth.

| Flood event | $R^2$ | MAE (meters) | NRMSE (%) | $F_Q$ (%) |
|---|---|---|---|---|
| **Original Model** | | | | |
| **Hurricane Ida** | 0.92 | 0.66 | 29 | 138 |
| **Transferability** | | | | |
| **Hurricane Isaias** | 0.77 | 1.44 | 80 | 322 |




| | | | | |
|---|---|---|---|---|
| **Hurricane Sandy** | 0.71 | 1.69 | 109 | 366 |
| **Hurricane Irene** | 0.8 | 1.19 | 43 | 113 |


These results demonstrated the model ability to generalize across different hurricanes within the
same watershed ($R^2 > 0.71$). With a MAE less than 1.69 m in all hurricanes, our model's
performance is consistent with Guo et al. (2021), demonstrating its capability for reasonable flood
depth estimates under hurricane conditions. However, when compared to the original model
performance on Hurricane Ida, the $R^2$ values and other metrics show weaker model performance
for the transferability to other hurricanes, suggesting reduced estimative accuracy, but not to the
extent that the model performance becomes unsatisfactory.

Figure 6 presents the flood estimations for all four events. In both Hurricanes Ida and Irene,

the model exhibited patterns of overestimation and underestimation across the study watershed.
For Hurricanes Isaias and Sandy, we primarily observed overestimations, which may be attributed
to their storm track locations. Furthermore, based on Figure 4, we mostly observe overestimation
in shallower locations and underestimation for deeper water levels at the stream gauges. This
pattern aligns with the southward total slope aspect, where the upper point of the river tends to
have shallower depths and the mainstream exhibits deeper water levels.

The model achieved an $R^2$ of 0.80 for Hurricane Irene, scoring 0.77 for Isaias and 0.71 for

Sandy. Based on table 2, Hurricanes Ida and Irene exhibited significant similarities in river water
levels and antecedent soil moisture. Given that river water level is the target variable and
antecedent soil moisture is a crucial feature, better model transferability for Hurricane Irene
compared to Hurricanes Isaias and Sandy are expected. The spatial relationship between storm
tracks and watershed locations also plays a part in the model performance. Both Hurricanes Ida



and Irene followed similar storm tracks, located on the watershed's eastern side within a
comparable distance range. In contrast, Irene tracked were on the west side of the watershed, and
Hurricane Sandy was further south from the watershed. The model input feature "distance to storm
track" played a significant role, contributing to better transferability to Hurricane Irene due to its
similarity with hurricane Ida. However, the ML model still demonstrated satisfactory performance
on Hurricane Sandy, suggesting some level of transferability, mainly because we incorporated a
wide array of pertinent flood influencing features. This sensitivity underscores the importance of
training ML models on diverse hurricane trajectories and proximity to improve the model
transferability. While the model performs well, the inconsistency of the success level of
transferability across flood events presents opportunities to incorporate additional features or
training approaches, enhancing the model robustness to different storm tracks relative to the
watershed.
The MAE values were higher for Hurricanes Sandy and Isaias, particularly when they were
farther away from the storm track. For instance, Hurricane Sandy had the highest MAE (1.69 m)
among the transferability cases, indicating larger estimation errors compared to the other
hurricanes. The model overestimated flood depths of Hurricanes Sandy and Isaias, while it
underestimated those during Hurricane Ida and Irene, likely due to their distance to the storm track.
Additionally, hurricanes Sandy and Isaias tend to yield higher $F_Q$ values. For example, Hurricane
Sandy had the highest $F_Q$ (366%), indicating larger discrepancies between the estimations and the
observed flood depths compared to Hurricanes Irene and Isaias.
These findings highlight the challenges of accurately hindcasting flood depths during more
severe hurricanes and underscore the importance of further refining the model to enhance its
performance in extreme events. Further investigations into the underlying features contributing to





these variations are crucial for improving flood hindcast models in the future. Insights gained from
this study can help develop transferable ML-based models that are computationally efficient for
flood hindcast.

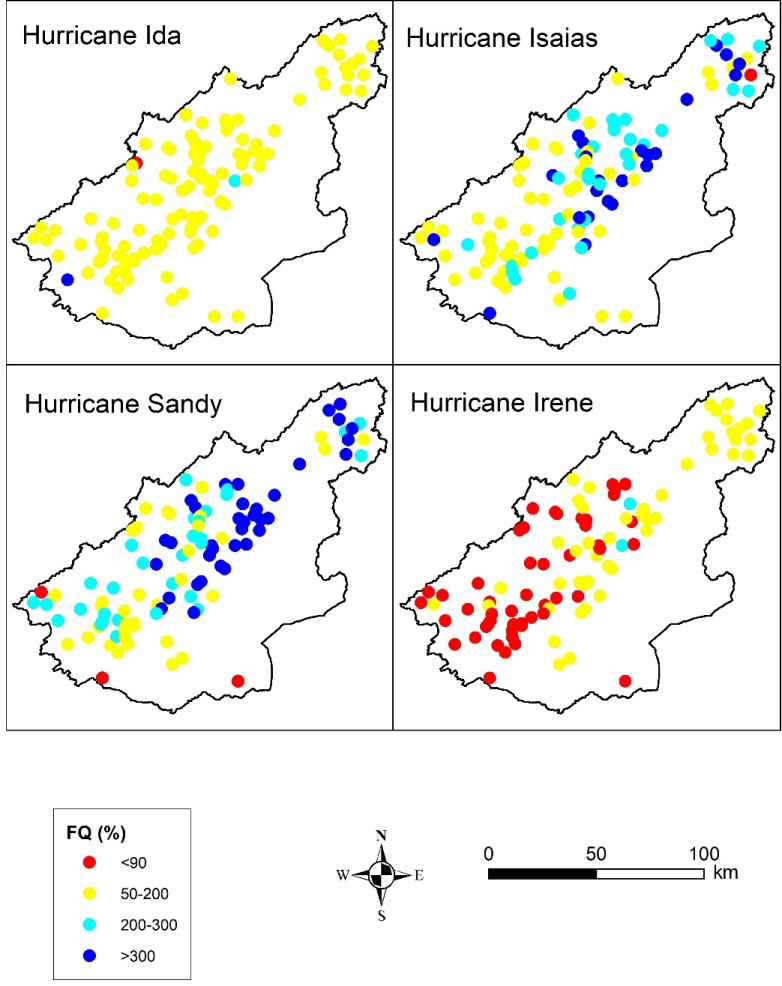


Figure 6: The ratio of estimated over observed flood depth ($F_Q$) for the four hurricanes.





### 4.4. Limitations and future research


While this study showed promising results about ML-based flood modeling, it is important to
acknowledge its limitations to identify areas for future research. One significant limitation is the
presence of inherent uncertainties in the model that can impact the accuracy of the estimations.
These uncertainties can stem from various sources, including the quality and accuracy of the input
data (features). For instance, relying solely on spatially aggregated values of features (mean and
maximum used in this study) may not adequately capture the complex characteristics of the upper
watershed. Future research should prioritize addressing these uncertainties by exploring alternative
data sources and methodologies. The ANN model was tuned using observed flood data and a
hyperparameter set was used as the optimal parameterization scenario. This deterministic approach
does not incorporate the uncertainty from model parameterization. Probabilistic models are needed
to address this uncertainty.
Furthermore, we did not have sub-daily data available for all our model features. Incorporating
sub-daily data can highly likely improve the model accuracy in capturing intra-daily variability
and flood dynamics, but it was not explored due to data constraints. Future research should
incorporate sub-daily data into flood depth hindcast models. A further limitation of this study
related to the time dimension is that wind events, storm surges, rainfall and overland flow
processes have different time signatures. Pluvial and storm surge flooding can be closely
coincident with the storm event, but river floodwaves may take much longer to arrive at a particular
location. The time lag between these processes was not considered in our ML model, which was
not dynamic in time and only hindcasted maximum river flood depths. Incorporating time-
variability of the features can better represent the time-varying nature of flood dynamics.



Another limitation of this study is the issue of bathymetry and the need for further analyses to
incorporate better data in coastal watersheds. However, using DEMs without added bathymetry is
not entirely inaccurate, as they can already include bathymetry information in regions where
LiDAR can penetrate beneath clear water surfaces, particularly in rivers with low suspended
sediment and turbidity. On the other hand, coastal floods confined within riverbanks may heavily
depend on the main channel slope, while extreme events leading to flooding outside the channel
banks follow the general slope of floodplains and this is easily represented by DEMs without
considering underwater bathymetry.
Additionally, we modeled flood depths across a large watershed (HUC6), whereby many
details may not be important. For small watersheds and specially urbanized ones, we emphasize
the importance of considering local factors such as sewer and drainage systems in flood depth
hindcast, where pluvial floods may be prevalent. However, obtaining comprehensive and accurate
data on sewer and drainage systems can be challenging due to availability, lack of quality and
confidentiality of the data, particularly at the desired spatial and temporal resolutions. Future
research should strive to improve the availability and accessibility of such data to enhance the
accuracy and reliability of flood depth hindcasting, especially in urban areas. In small urban
watersheds, other details such as land management practices and other local features can also be
important for flood depth hindcasting and should be incorporated in the ML-based model.
This study primarily focused on hindcasting maximum flood depths and did not consider other
important flood characteristics, such as flood duration, frequency, and extent, all of which are
important for loss estimates, decision making and risk management (Ahmadisharaf and Kalyanapu
2019; Kreibich et al. 2009; Merz et al. 2010; H. Qi and Altinakar 2011b; 2011a; 2012). To gain a
fuller picture of flood hazards, future research should aim to develop ML models that can hindcast

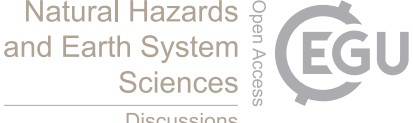

these additional flood characteristics. We also focused on river flood depths and did not hindcast
inundation on floodplains. Developing ML-based models that can satisfactorily hindcast out-of-
channel flood depths should be a focus of future research; the transferability of ML-based models
for such estimations should be also evaluated. High water marks (HWMs) can be used to train the
model for such hindcasting. However, HWMs are subject to large uncertainties (Schubert et al.
2022). Therefore, one challenge in developing models that hindcast flood depths over floodplains
is the availability of reliable observations. Satellite-based observations are also often limited to
flood status data; flood depths cannot be estimated using these types of datasets. Newly launched
satellites, such as the Surface Water and Ocean Topography (SWOT) mission, can provide
additional data for such estimations.
As part of future work, it is also essential to consider the sensitivity of stream gauges to changes
in flow once water exceeds bankfull levels. This is significant as water height changes at a slower
rate beyond bankfull due to the compound channel shape. Wide floodplains can lead to similar
stage elevations for quite different flow conditions. This sensitivity assessment can offer insights
about whether water levels can be estimated once flood conditions are established, which has
implications for the model transferability across events.
We recommend that future work compares the performance of our ML-based model to
traditional physically-based and morphologic-based models using the same datasets. By evaluating
the performance, generalizability, and computational efficiency of our ML-based model versus
these traditional modeling approaches, we will be able to better validate the strengths of our data-
driven methodology. Detailed error analyses between the approaches can also reveal insights into
where additional physics knowledge needs to be incorporated into the ML-based model structure
and training to improve performance.



Thus, although we found ML-based models are transferable across flood events when informed
by relevant physical features at meaningful locations, there are still several areas that require
further investigations. By addressing these limitations, future research can corroborate our findings
about the performance and transferability of ML-based models in estimating maximum flood
depths as computationally-efficient modeling frameworks.

**5.    Summary and conclusions**

This paper developed an ML-based model for hindcast maximum flood depths to address two
major limitations of past research in applying ML models for flood estimations: solely predicting
flood status (classification-based models) and debate on the transferability of these models across
events. We used ANN to hindcast maximum flood depths over an event on a coastal watershed,
which is affected by fluvial and tidal floods. The model was informed by underlying physical flood
processes, represented through a set of features (geographic location, topographic, climatic, land
surface, hydrologic, hydrodynamic and soil). Unlike previous applications of ML algorithms, our
model estimated flood depths by accounting for the spatial distribution of the processes through
considering both local contributions (at a given location) and those from the upstream watersheds.
We demonstrated the model on a HUC6 watershed, Lower Hudson Watershed, in the Northeastern
United States and evaluated its transferability across major flood events—Hurricanes Ida, Sandy,
Irene and Isaias. Feature selection techniques were used to identify the most influential features
for flood hindcast. Hyperparameter optimization was performed to fine-tune the ML model, and
its performance was evaluated using various metrics. The results showed that the model performed
satisfactorily in estimating maximum flood depths for the original event, Hurricane Ida ($R^2$= 0.92,
MAE= 0.66, NRMSE= 29%, and $F_Q$= 139%). The model transferability (i.e., applying the
validated model as is without any additional parameter tuning) within the same watershed against





three other events showed that the developed model was promising in the estimations ($R^2 > 0.71$,
MAE< 1.69, NRMSE < 109%, and $F_Q$< 366%). This showed the model ability to capture complex
relationships between the maximum flood depth and pertinent features beyond what it was
originally trained for. Future research is needed to further evaluate the transferability of ML
models across events and watersheds with different drainage areas for flood depth estimations.
**Author contribution**
**MP:** Data curation, Formal analysis, Investigation, Methodology, Software, Validation,
Visualization, Writing – original draft preparation; **EA:** Conceptualization, Methodology, Funding
acquisition, Project administration, Supervision, Writing – review & editing; **BN**: Methodology,
Writing – review & editing; **EC**: Visualization, Writing – review & editing.
**Code availability**
The ML codes can be shared upon request.
**Data availability**
All the data are public domain and can be acquired from online repositories.
**Competing interests**
The contact author has declared that none of the authors has any competing interests
**Acknowledgements**
This study was partially supported through a research grant by United States' National Science
Foundation (award number 2203180). We thank Paul Bates for the detailed review and fruitful
comments on this manuscript.



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

Limitations of the Diffusive Approximation of the 2-d Shallow Water Equations for Flood

Simulation in Urban and Rural Areas." *Applied Numerical Mathematics*, New Trends in

Numerical Analysis: Theory, Methods, Algorithms and Applications (NETNA 2015), 116

(June): 141–56. https://doi.org/10.1016/j.apnum.2016.07.003.

Davenport, Frances V., Marshall Burke, and Noah S. Diffenbaugh. 2021. "Contribution of

Historical Precipitation Change to US Flood Damages." *Proceedings of the National*

*Academy of Sciences* 118 (4): e2017524118. https://doi.org/10.1073/pnas.2017524118.


Dawson, C. W., R. J. Abrahart, A. Y. Shamseldin, and R. L. Wilby. 2006. "Flood Estimation at

Ungauged Sites Using Artificial Neural Networks." *Journal of Hydrology* 319 (1): 391–

409. https://doi.org/10.1016/j.jhydrol.2005.07.032.

Elkhrachy, Ismail. 2022. "Flash Flood Water Depth Estimation Using SAR Images, Digital

Elevation Models, and Machine Learning Algorithms." *Remote Sensing* 14 (3): 440.

https://doi.org/10.3390/rs14030440.

Fernández-Pato, Javier, Daniel Caviedes-Voullième, and Pilar García-Navarro. 2016.

"Rainfall/Runoff Simulation with 2D Full Shallow Water Equations: Sensitivity Analysis

and Calibration of Infiltration Parameters." *Journal of Hydrology* 536 (May): 496–513.

https://doi.org/10.1016/j.jhydrol.2016.03.021.

Galloway, Gerald E, Allison Reilly, Sung Ryoo, Anjanette Riley, Maggie Haslam, Sam Brody,

Wesley Highfield, Joshua Goldstein, Jayton Rainey, and Sherry Parker,. 2018. "Urban-

Flooding-Report-Online.Pdf." THE GROWING THREAT OF URBAN FLOODING: A

NATIONAL        CHALLENGE.        2018.        https://today.tamu.edu/wp-

content/uploads/sites/4/2018/11/Urban-flooding-report-online.pdf.

Gray W. Brunner. 2016. "HEC-RAS, River Analysis System Hydraulic Reference Manual."

February  2016.  https://www.hec.usace.army.mil/software/hec-ras/documentation/HEC-

RAS%205.0%20Reference%20Manual.pdf.

Gudiyangada Nachappa, Thimmaiah, Sepideh Tavakkoli Piralilou, Khalil Gholamnia, Omid

Ghorbanzadeh, Omid Rahmati, and Thomas Blaschke. 2020. "Flood Susceptibility

Mapping with Machine Learning, Multi-Criteria Decision Analysis and Ensemble Using

Dempster    Shafer    Theory."    *Journal    of    Hydrology*    590    (November):    125275.

https://doi.org/10.1016/j.jhydrol.2020.125275.



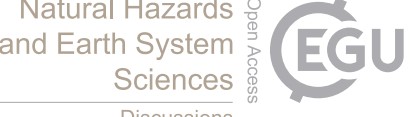

Guo, Zifeng, João P. Leitão, Nuno E. Simões, and Vahid Moosavi. 2021. "Data-Driven Flood

Emulation: Speeding up Urban Flood Predictions by Deep Convolutional Neural

Networks." *Journal of Flood Risk Management* 14 (1): e12684.

https://doi.org/10.1111/jfr3.12684.

Hashmi, Farukh. 2020. "How to Tune Hyperparameters Using Random Search CV in Python."

*Thinking Neuron* (blog). September 10, 2020. https://thinkingneuron.com/how-to-tune-

hyperparameters-using-random-search-cv-in-python/.

Hemmati, Mona, Bruce R. Ellingwood, and Hussam N. Mahmoud. 2020. "The Role of Urban

Growth in Resilience of Communities Under Flood Risk." *Earth's Future* 8 (3).

https://doi.org/10.1029/2019EF001382.

Hino, Miyuki, and Earthea Nance. 2021. "Five Ways to Ensure Flood-Risk Research Helps the

Most Vulnerable." *Nature* 595 (7865): 27–29. https://doi.org/10.1038/d41586-021-01750-

0.

Hosseini, Farzaneh Sajedi, Bahram Choubin, Amir Mosavi, Narjes Nabipour, Shahaboddin

Shamshirband, Hamid Darabi, and Ali Torabi Haghighi. 2020. "Flash-Flood Hazard

Assessment Using Ensembles and Bayesian-Based Machine Learning Models: Application

of the Simulated Annealing Feature Selection Method." *Science of The Total Environment*

711 (April): 135161. https://doi.org/10.1016/j.scitotenv.2019.135161.

Hosseiny, Hossein, Foad Nazari, Virginia Smith, and C. Nataraj. 2020. "A Framework for

Modeling Flood Depth Using a Hybrid of Hydraulics and Machine Learning." *Scientific*

*Reports* 10 (1): 8222. https://doi.org/10.1038/s41598-020-65232-5.

Hu, Anson, and Ibrahim Demir. 2021. "Real-Time Flood Mapping on Client-Side Web Systems

Using HAND Model." *Hydrology* 8 (2): 65. https://doi.org/10.3390/hydrology8020065.



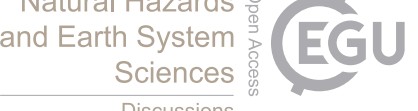

Huang, Faming, Siyu Tao, Deying Li, Zhipeng Lian, Filippo Catani, Jinsong Huang, Kailong Li,

and Chuhong Zhang. 2022. "Landslide Susceptibility Prediction Considering

Neighborhood Characteristics of Landslide Spatial Datasets and Hydrological Slope Units

Using Remote Sensing and GIS Technologies." *Remote Sensing* 14 (18): 4436.

https://doi.org/10.3390/rs14184436.

Jafarzadegan, Keighobad, and Venkatesh Merwade. 2019. "Probabilistic Floodplain Mapping

Using HAND-Based Statistical Approach." *Geomorphology* 324 (January): 48–61.

https://doi.org/10.1016/j.geomorph.2018.09.024.

Jafarzadegan, Keighobad, Hamid Moradkhani, Florian Pappenberger, Hamed Moftakhari, Paul

Bates, Peyman Abbaszadeh, Reza Marsooli, et al. 2023. "Recent Advances and New

Frontiers in Riverine and Coastal Flood Modeling." *Reviews of Geophysics* 61 (2):

e2022RG000788. https://doi.org/10.1029/2022RG000788.

Joseph, V. Roshan. 2022. "Optimal Ratio for Data Splitting." *Statistical Analysis and Data*

*Mining: The ASA Data Science Journal* 15 (4): 531–38.

https://doi.org/10.1002/sam.11583.

Kalyanapu, Alfred J., Siddharth Shankar, Eric R. Pardyjak, David R. Judi, and Steven J. Burian.

2011. "Assessment of GPU Computational Enhancement to a 2D Flood Model."

*Environmental Modelling & Software* 26 (8): 1009–16.

https://doi.org/10.1016/j.envsoft.2011.02.014.

Khosravi, Khabat, Binh Thai Pham, Kamran Chapi, Ataollah Shirzadi, Himan Shahabi, Inge

Revhaug, Indra Prakash, and Dieu Tien Bui. 2018. "A Comparative Assessment of

Decision Trees Algorithms for Flash Flood Susceptibility Modeling at Haraz Watershed,



938 Northern Iran." *Science of The Total Environment* 627 (June): 744–55.

939 https://doi.org/10.1016/j.scitotenv.2018.01.266.

940 Kim, Sooyoul, Yoshiharu Matsumi, Shunqi Pan, and Hajime Mase. 2016. "A Real-Time Forecast

941 Model Using Artificial Neural Network for after-Runner Storm Surges on the Tottori

942 Coast, Japan." *Ocean Engineering* 122 (August): 44–53.

943 https://doi.org/10.1016/j.oceaneng.2016.06.017.

944 Kratzert, Frederik, Daniel Klotz, Mathew Herrnegger, Alden K. Sampson, Sepp Hochreiter, and

945 Grey S. Nearing. 2019. "Toward Improved Predictions in Ungauged Basins: Exploiting the

946 Power of Machine Learning." *Water Resources Research* 55 (12): 11344–54.

947 https://doi.org/10.1029/2019WR026065.

948 Kreibich, H., K. Piroth, I. Seifert, H. Maiwald, U. Kunert, J. Schwarz, B. Merz, and A. H. Thieken.

949 2009. "Is Flow Velocity a Significant Parameter in Flood Damage Modelling?" *Natural*

950 *Hazards and Earth System Sciences* 9 (5): 1679–92. https://doi.org/10.5194/nhess-9-1679-

951 2009.

952 Kulp, Scott A., and Benjamin H. Strauss. 2019. "New Elevation Data Triple Estimates of Global

953 Vulnerability to Sea-Level Rise and Coastal Flooding." *Nature Communications* 10 (1):

954 4844. https://doi.org/10.1038/s41467-019-12808-z.

955 Kundzewicz, ZW, Buda Su, Yanjun Wang, Jun Xia, Jinlong Huang, and Tong Jiang. 2019. "Flood

956 Risk and Its Reduction in China." *Advances in Water Resources* 130 (August): 37–45.

957 https://doi.org/10.1016/j.advwatres.2019.05.020.

958 Latto, Andy, Andrew Hagen, and Robbie Berg. 2021. "Tropical Cyclone Report - HURRICANE

959 ISAIAS (AL092020)." National Hurricane Center. June 11, 2021.

960 https://www.nhc.noaa.gov/data/tcr/AL092020_Isaias.pdf.

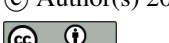

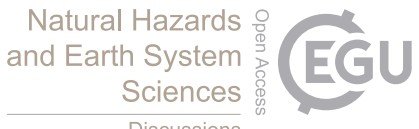

Lee, Deuk-Hwan, Yun-Tae Kim, and Seung-Rae Lee. 2020. "Shallow Landslide Susceptibility

Models Based on Artificial Neural Networks Considering the Factor Selection Method and

Various Non-Linear Activation Functions." *Remote Sensing* 12 (7): 1194.

https://doi.org/10.3390/rs12071194.

Lixion A., Avila, and John Cangialosi. 2013. "Tropical Cyclone Report -  Hurricane Irene

(AL092011)." National Hurricane Center. April 11, 2013.

https://www.nhc.noaa.gov/data/tcr/AL092011_Irene.pdf.

Löwe, Roland, Julian Böhm, David Getreuer Jensen, Jorge Leandro, and Søren Højmark

Rasmussen. 2021. "U-FLOOD – Topographic Deep Learning for Predicting Urban Pluvial

Flood Water Depth." *Journal of Hydrology* 603 (December): 126898.

https://doi.org/10.1016/j.jhydrol.2021.126898.

Lundberg, Scott, and Su-In Lee. 2017. "A Unified Approach to Interpreting Model Predictions."

arXiv. http://arxiv.org/abs/1705.07874.

Maier, Holger R., Stefano Galelli, Saman Razavi, Andrea Castelletti, Andrea Rizzoli, Ioannis N.

Athanasiadis, Miquel Sànchez-Marrè, Marco Acutis, Wenyan Wu, and Greer B.

Humphrey. 2023. "Exploding the Myths: An Introduction to Artificial Neural Networks

for Prediction and Forecasting." *Environmental Modelling & Software* 167 (September):

105776. https://doi.org/10.1016/j.envsoft.2023.105776.

McCulloch, Warren S., and Walter Pitts. 1943. "A Logical Calculus of the Ideas Immanent in

Nervous Activity." *The Bulletin of Mathematical Biophysics* 5 (4): 115–33.

https://doi.org/10.1007/BF02478259.



Merz, B, Heidi Kreibich, R Schwarze, and Annette Thieken. 2010. "Review Article" Assessment

of Economic Flood Damage"." *Natural Hazards and Earth System Sciences* 10: 1697–

1724. https://doi.org/10.5194/nhess-10-1697-2010.

Ming, Xiaodong, Qiuhua Liang, Xilin Xia, Dingmin Li, and Hayley J. Fowler. 2020. "Real-Time

Flood Forecasting Based on a High-Performance 2-D Hydrodynamic Model and

Numerical Weather Predictions." *Water Resources Research* 56 (7): e2019WR025583.

https://doi.org/10.1029/2019WR025583.

Mishra, Ashok, Sourav Mukherjee, Bruno Merz, Vijay P. Singh, Daniel B. Wright, Villarini

Gabriele, Subir Paul, et al. 2022. "An Overview of Flood Concepts, Challenges, and Future

Directions."     *Journal     of     Hydrologic     Engineering     27     (6).*

https://ascelibrary.org/doi/full/10.1061/(ASCE)HE.1943-5584.0002164.

Mosavi, Amir, Pinar Ozturk, and Kwok-wing Chau. 2018. "Flood Prediction Using Machine

Learning     Models:     Literature     Review."     *Water*     10     (11):     1536.

https://doi.org/10.3390/w10111536.

National Academies of Sciences, Engineering, and Medicine. 2019. *Framing the Challenge of*

*Urban Flooding in the United States*. Washington, DC: The National Academies Press.

https://doi.org/10.17226/25381.

National     Hurricane     Center.     2022.     "National     Hurricane     Center."     2022.

https://www.nhc.noaa.gov/index.shtml.

Nguyen, Quang Hung, Hai-Bang Ly, Lanh Si Ho, Nadhir Al-Ansari, Hiep Van Le, Van Quan Tran,

Indra Prakash, and Binh Thai Pham. 2021. "Influence of Data Splitting on Performance of

Machine Learning Models in Prediction of Shear Strength of Soil." *Mathematical*



*Problems in Engineering* 2021 (February): e4832864. https://doi.org/10.1155/2021/4832864.

"NOAA Tides & Currents." 2023. CO-OPS Map - NOAA Tides & Currents. 2023. https://tidesandcurrents.noaa.gov/map/index.html.

NOAA's NCEI. 2022. "Data Search | National Centers for Environmental Information (NCEI)." 2022. https://www.ncei.noaa.gov/access/search/data-search/local-climatological-data.

Pham, Binh Thai, Chinh Luu, Tran Van Phong, Phan Trong Trinh, Ataollah Shirzadi, Somayeh Renoud, Shahrokh Asadi, Hiep Van Le, Jason von Meding, and John J. Clague. 2021. "Can Deep Learning Algorithms Outperform Benchmark Machine Learning Algorithms in Flood Susceptibility Modeling?" *Journal of Hydrology* 592 (January): 125615. https://doi.org/10.1016/j.jhydrol.2020.125615.

Pradhan, Biswajeet. 2009. "Journal of Spatial Hydrology Vol.9, No.2 Fall 2009."

Qi, Honghai, and Mustafa S. Altinakar. 2011a. "A Conceptual Framework of Agricultural Land Use Planning with BMP for Integrated Watershed Management." *Journal of Environmental Management* 92 (1): 149–55. https://doi.org/10.1016/j.jenvman.2010.08.023.

———. 2011b. "Vegetation Buffer Strips Design Using an Optimization Approach for Non-Point Source Pollutant Control of an Agricultural Watershed." *Water Resources Management* 25 (2): 565–78. https://doi.org/10.1007/s11269-010-9714-9.

———. 2012. "GIS-Based Decision Support System for Dam Break Flood Management under Uncertainty with Two-Dimensional Numerical Simulations." *Journal of Water Resources Planning and Management* 138 (4): 334–41. https://doi.org/10.1061/(ASCE)WR.1943-5452.0000192.



Qi, Wenchao, Chao Ma, Hongshi Xu, Zifan Chen, Kai Zhao, and Hao Han. 2021. "A Review on

Applications of Urban Flood Models in Flood Mitigation Strategies." *Natural Hazards* 108

(1): 31–62. https://doi.org/10.1007/s11069-021-04715-8.

Rafiei-Sardooi, Elham, Ali Azareh, Bahram Choubin, Amir H. Mosavi, and John J. Clague. 2021.

"Evaluating Urban Flood Risk Using Hybrid Method of TOPSIS and Machine Learning."

*International Journal of Disaster Risk Reduction* 66 (December): 102614.

https://doi.org/10.1016/j.ijdrr.2021.102614.

Rahmati, Omid, Hamid Reza Pourghasemi, and Hossein Zeinivand. 2016. "Flood Susceptibility

Mapping Using Frequency Ratio and Weights-of-Evidence Models in the Golastan

Province, Iran." *Geocarto International* 31 (1): 42–70.

https://doi.org/10.1080/10106049.2015.1041559.

Reckien, Diana. 2018. "What Is in an Index? Construction Method, Data Metric, and Weighting

Scheme Determine the Outcome of Composite Social Vulnerability Indices in New York

City." *Regional Environmental Change* 18 (5): 1439–51. https://doi.org/10.1007/s10113-

017-1273-7.

Rennó, Camilo Daleles, Antonio Donato Nobre, Luz Adriana Cuartas, João Vianei Soares, Martin

G. Hodnett, Javier Tomasella, and Maarten J. Waterloo. 2008. "HAND, a New Terrain

Descriptor Using SRTM-DEM: Mapping Terra-Firme Rainforest Environments in

Amazonia." *Remote Sensing of Environment* 112 (9): 3469–81.

https://doi.org/10.1016/j.rse.2008.03.018.

Rezaie, Fatemeh, Mahdi Panahi, Sayed M. Bateni, Changhyun Jun, Christopher M. U. Neale, and

Saro Lee. 2022. "Novel Hybrid Models by Coupling Support Vector Regression (SVR)



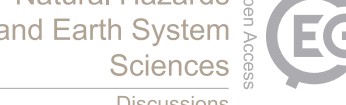

with Meta-Heuristic Algorithms (WOA and GWO) for Flood Susceptibility Mapping."

*Natural Hazards* 114 (2): 1247–83. https://doi.org/10.1007/s11069-022-05424-6.

Rumelhart, David E., James L. McClelland, and PDP Research Group. 1986. *Parallel Distributed*

*Processing: Explorations in the Microstructure of Cognition: Foundations*. The MIT

Press. https://doi.org/10.7551/mitpress/5236.001.0001.

Salvati, Aryan, Alireza Moghaddam Nia, Ali Salajegheh, Kayvan Ghaderi, Dawood Talebpour

Asl, Nadhir Al-Ansari, Feridon Solaimani, and John J. Clague. 2023. "Flood Susceptibility

Mapping Using Support Vector Regression and Hyper-Parameter Optimization." *Journal*

*of Flood Risk Management* n/a (n/a): e12920. https://doi.org/10.1111/jfr3.12920.

Schubert, Jochen E., Adam Luke, Amir AghaKouchak, and Brett F. Sanders. 2022. "A Framework

for Mechanistic Flood Inundation Forecasting at the Metropolitan Scale." *Water Resources*

*Research* 58 (10): e2021WR031279. https://doi.org/10.1029/2021WR031279.

Schubert, Jochen E., and Brett F. Sanders. 2012. "Building Treatments for Urban Flood Inundation

Models and Implications for Predictive Skill and Modeling Efficiency." *Advances in Water*

*Resources* 41 (June): 49–64. https://doi.org/10.1016/j.advwatres.2012.02.012.

Sridhar, Venkataramana, Syed Azhar Ali, and David J. Sample. 2021. "Systems Analysis of

Coupled Natural and Human Processes in the Mekong River Basin." *Hydrology* 8 (3): 140.

https://doi.org/10.3390/hydrology8030140.

Stow, Craig A., Chris Roessler, Mark E. Borsuk, James D. Bowen, and Kenneth H. Reckhow.

2003. "Comparison of Estuarine Water Quality Models for Total Maximum Daily Load

Development in Neuse River Estuary." *Journal of Water Resources Planning and*

*Management* 129 (4): 307–14. https://doi.org/10.1061/(ASCE)0733-

9496(2003)129:4(307).

Natural Hazards
and Earth System
Sun, Deliang, Jiahui Xu, Haijia Wen, and Yue Wang. 2020. "An Optimized Random Forest Model

and Its Generalization Ability in Landslide Susceptibility Mapping: Application in Two

Areas of Three Gorges Reservoir, China." *Journal of Earth Science* 31 (6): 1068–86.

https://doi.org/10.1007/s12583-020-1072-9.

USGS. 2022. "TNM Download V2." 2022. https://apps.nationalmap.gov/downloader/.

Viglione, Alberto, Giuliano Di Baldassarre, Luigia Brandimarte, Linda Kuil, Gemma Carr, José

Luis Salinas, Anna Scolobig, and Günter Blöschl. 2014. "Insights from Socio-Hydrology

Modelling on Dealing with Flood Risk – Roles of Collective Memory, Risk-Taking

Attitude and Trust." *Journal of Hydrology*, Creating Partnerships Between Hydrology and

Social Science: A Priority for Progress, 518 (October): 71–82.

https://doi.org/10.1016/j.jhydrol.2014.01.018.

Wan Jaafar, Wan Zurina, and Dawei Han. 2012. "Uncertainty in Index Flood Modelling Due to

Calibration Data Sizes." *Hydrological Processes* 26 (2): 189–201.

https://doi.org/10.1002/hyp.8135.

Wang, Jie, Qiuhong Tang, Xiaobo Yun, Aifang Chen, Siao Sun, and Dai Yamazaki. 2022. "Flood

Inundation in the Lancang-Mekong River Basin: Assessing the Role of Summer

Monsoon." *Journal of Hydrology* 612 (September): 128075.

https://doi.org/10.1016/j.jhydrol.2022.128075.

Wang, Zhaoli, Chengguang Lai, Xiaohong Chen, Bing Yang, Shiwei Zhao, and Xiaoyan Bai.

2015. "Flood Hazard Risk Assessment Model Based on Random Forest." *Journal of*

*Hydrology* 527 (August): 1130–41. https://doi.org/10.1016/j.jhydrol.2015.06.008.

Wing, Oliver E. J., William Lehman, Paul D. Bates, Christopher C. Sampson, Niall Quinn, Andrew

1094        M. Smith, Jeffrey C. Neal, Jeremy R. Porter, and Carolyn Kousky. 2022. "Inequitable



Patterns of US Flood Risk in the Anthropocene." *Nature Climate Change* 12 (2): 156–62.

https://doi.org/10.1038/s41558-021-01265-6.

Youssef, Ahmed M., Biswajeet Pradhan, Abhirup Dikshit, and Ali M. Mahdi. 2022. "Comparative

Study of Convolutional Neural Network (CNN) and Support Vector Machine (SVM) for

Flood Susceptibility Mapping: A Case Study at Ras Gharib, Red Sea, Egypt." *Geocarto*

*International* 37 (26): 11088–115. https://doi.org/10.1080/10106049.2022.2046866.

Zahura, Faria T., Jonathan L. Goodall, Jeffrey M. Sadler, Yawen Shen, Mohamed M. Morsy, and

Madhur Behl. 2020. "Training Machine Learning Surrogate Models From a High-Fidelity

Physics-Based Model: Application for Real-Time Street-Scale Flood Prediction in an

Urban Coastal Community." *Water Resources Research* 56 (10).

https://doi.org/10.1029/2019WR027038.

Zhang, Fang, Xiaolin Zhu, and Desheng Liu. 2014. "Blending MODIS and Landsat Images for

Urban Flood Mapping." *International Journal of Remote Sensing* 35 (9): 3237–53.

https://doi.org/10.1080/01431161.2014.903351.

Zhao, Gang, Bo Pang, Zongxue Xu, Lizhuang Cui, Jingjing Wang, Depeng Zuo, and Dingzhi

Peng. 2021. "Improving Urban Flood Susceptibility Mapping Using Transfer Learning."

*Journal of Hydrology* 602 (November): 126777.

https://doi.org/10.1016/j.jhydrol.2021.126777.

Zhao, Gang, Bo Pang, Zongxue Xu, Dingzhi Peng, and Depeng Zuo. 2020. "Urban Flood

Susceptibility Assessment Based on Convolutional Neural Networks." *Journal of*

*Hydrology* 590 (November): 125235. https://doi.org/10.1016/j.jhydrol.2020.125235.

Zheng, Xing, David G. Tarboton, David R. Maidment, Yan Y. Liu, and Paola Passalacqua. 2018.

"River Channel Geometry and Rating Curve Estimation Using Height above the Nearest



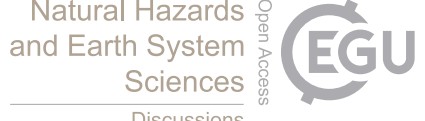

Drainage." *JAWRA Journal of the American Water Resources Association* 54 (4): 785–

806. https://doi.org/10.1111/1752-1688.12661.

Zhu, D., Q. Ren, Y. Xuan, Y. Chen, and I. D. Cluckie. 2013. "An Effective Depression Filling

Algorithm for DEM-Based 2-D Surface Flow Modelling." *Hydrology and Earth System*

*Sciences* 17 (2): 495–505. https://doi.org/10.5194/hess-17-495-2013.

Zhu, Jun-Jie, Meiqi Yang, and Zhiyong Jason Ren. 2023. "Machine Learning in Environmental

Research: Common Pitfalls and Best Practices." *Environmental Science & Technology*,

June. https://doi.org/10.1021/acs.est.3c00026.
