# Peer review of "Transferability of machine learning-based modeling frameworks across flood events for hindcasting maximum river flood depths in coastal watersheds"

_Natural Hazards and Earth System Sciences, 2023_

## Author Comment (AC1)

**Manuscript Reference:** NHESS-2023-152

**Manuscript Title:** Transferability of machine learning-based modeling frameworks across flood events for hindcasting maximum river flood depths in coastal watersheds

**Summary:** This paper explores the application of ML to hindcasting maximum river flood depths across events when informed by the spatial distribution of pertinent features and underlying physical processes in coastal watersheds. Trained within the same watershed, the ML model was then transferred in time to predict out-of-sample events for three major storms events. Acceptable performance was noted at 100+ gauge stations in the domain.

**General Comments:**

1. The content of the literature review (and motivation overall) was helpful and sound (applicable references to previous studies), highlighting gaps in the current field and therefore valid contributions that would be efficient for those wanting to hindcast flood depths for tropical storms without the need to use a hydrologic model.

2. Scientific Significance: An ML model that moves past simple categorical prediction of expecting a flood or not in a watershed to estimate actual flood depths at stream gauges; and an approach that allows for hindcasting of storm events in the watershed not previously seen or trained by the model are two main contributions of this work. While it is necessary for communities to understand expected depths along flood plains and other areas in a watershed, the study takes a key first step towards establishing a methodology for maximum flood depth estimation at point locations in a stream with an ML model that requires input features that could be harnessed from available datasets. Though retained to

point estimations of flood depths for the study in a single HUC-06 watershed, the methodology permits application/implementation to watersheds of various size and locations. The study addresses scientific questions within the scope of NHESS.

3. Scientific Quality: The PCA and SHAP analyses for assessing the importance of features on flood depth estimation will be very useful for the hydrological community. Of course these features will vary per watershed and future work may further inform potential important features currently not included that will improve performance – but, feature analyses such as these coupled with ML models are what help to make the black-box of ML models interpretable and hydrologists piece together valuable information on the physics behind flood events in a watershed without the need for setting up and running hydrologic models. Overfitting is always a concern for ML models but some steps to reduce "redundant" variables by computing correlation among variables was taken to help.

4. Presentation Quality: The overall presentation of the manuscript was well-organized and easy to follow – from the perspectives of both hydrology and ML. Figures and tables are easy to understand. There are minor grammar edits needed, but language otherwise was precise and fluent.

**Response:** We thank the reviewer for the constructive comments. Your review has helped us improve the manuscript quality. Detailed responses are provided to your questions. Blue text shows our response and black text shows your comments.

In the revised manuscript, we have added new features, conducted new analyses to evaluate the feature importance, performed new modeling to improve the performance and expanded our discussion to provide more insights about our results.

**Minor Suggestions/Technical corrections:**

1. Considering the mentioned overestimation of shallow flood depths and underestimation of high flood depths, perhaps a median metric would be a more robust test of performance (than MAE)?

**Response:** We have now included the median absolute error (MDAE) as an additional metric to evaluate our model performance. This enhancement aims to present a balanced view of the model accuracy across different flood depths. These new results are presented in Lines 577-582:

"The model demonstrated an excellent performance on the training dataset ($R^2$ = 0.94, MAE = 0.64 m, MDAE = 0.44 m, and NRMSE = 24%). On the test dataset, the model achieved an $R^2$ of 0.91, the MAE of 0.77 m, MDAE was 0.42 m, and the NRMSE was 28%, further suggesting the satisfactory performance by the model. The training history plot showed that the model performance improved with each epoch during training, indicating that the model was learning from the data. The model training process stopped at epoch 87 due to early stopping."

In Lines 621 to 635:

"The transferability of the trained and tested model (against Hurricane Ida) was examined by applying it to three other events within the same watershed. Table 4 summarizes the evaluation metrics for the three hurricanes.

Table 4. Model performance across in historical flood events. MAE: mean absolute error; MDAE: Median Absolute Error; RMSE: root mean square error; FQ: ratio of estimated over observed maximum flood depth.

| Flood event | $R^2$ | MAE (meters) | MDAE (meters) | NRMSE (%) | $F_Q$ (%) |
|---|---|---|---|---|---|
| **Original model** | | | | | |
| Hurricane Ida | 0.94 | 0.64 | 0.45 | 24.1 | 138.1 |
| **Transferability** | | | | | |
| Hurricane Isaias | 0.73 | 1.54 | 0.85 | 86.3 | 325.6 |
| Hurricane Sandy | 0.70 | 1.71 | 1.78 | 109.2 | 370.2 |
| Hurricane Irene | 0.85 | 1.12 | 0.85 | 36.7 | 112.6 |

These results demonstrated the model ability to transfer across different hurricanes within the same watershed (R2>0.70). With an MAE less than 1.71 m in all hurricanes, our model performance is consistent with the CNN model of Guo et al. (2021), demonstrating its capability for satisfactory flood depth estimates. However, when compared to the original model performance on Hurricane Ida, the R2 values and other metrics show weaker model performance for the transferability to other hurricanes, suggesting reduced estimative accuracy, but not to the extent that the model performance becomes unsatisfactory."

2. Also, given the vast difference noted between the NRMSE and the simpler FQ ratios, it would be interesting to see the scatter plot of simulated vs observed max flood depths for the storm events. This may even shed some light (on further understanding) where performance is good and not so good to help improve max flood depth estimation.

**Response:** We generated scatter plots of the estimated versus observed maximum flood depths for the events (Figures 6 and 8). These plots illustrate the model performance across the spectrum of flood depths, identifying areas of both strength and potential improvement in predicting maximum flood depths Lines 585 to 595:

"Figure 6 shows the performance of the ML model in hindcasting maximum water depths at stream gauges, comparing estimated values against observed values for both training and testing datasets. In the training phase (Figure 6a), points are clustered along the identity line, but tend to underestimate large water depths. This pattern suggested that the model learned the training data well, especially for smaller water depths, but did not fully capture the behavior that leads to the larger water depths. The underestimation of high values is expected due to the lower number of observations. The test data (Figure 6b) revealed a similar pattern of underestimation towards higher values; this can be since the number of observed high water depths is small."

[Figure]

Figure 6. Scatter plots of estimated vs observed maximum water depths for: (a) train and (b) test data. The identity line represents a perfect match between the estimated and observed values.

In Lines 636 to 644:

"Figure 8 shows the relationship between observed and estimated maximum water depths for the four storm events. Most observed water depths for the hurricanes were low. For all four events, the data points suggested that the model tends to underestimate the high water depths and overestimate the low water depths (Figure 8). The plots for Hurricanes Sandy and Irene show a

more dispersed set of points, suggesting a wider variance in the model estimates compared to the observations. This implied that the model is less accurate in capturing the flood dynamics of these events or that these events have unique characteristics that are not fully learned by the ML model.

[Figure]

Figure 8. Scatter plots of estimated vs observed flood depth for the four hurricanes.

3. The ML model caters to multiple types of flooding (both fluvial and coastal) – as the intention was the hindcast depths at stream gauges using these events and using the same model to do locations where inland flooding is more likely versus coastal flooding (and vice versa); is there a trend or pattern noted for performance in areas susceptible to fluvial versus coastal flooding or storm surge? Finding that Hurricanes Isaias and Sandy were overestimating flood depths further from the storm tracks (tracks that were further from coastal locations for both cases) – is there a chance that it is skewed towards one flood type?

**Response:** Our analyses suggested a pattern where the model overestimates flood depths in areas farther from the storm track, notably for Hurricanes Isaias and Sandy. This observation indicated that the model performs better in certain flood types (fluvial vs storm surge). It also suggested that having separate models for different flood types or training a single model on diverse flood event data, can enhance the performance by accommodating the specific characteristics of each flood type. The discrepancies in our predictions, especially the overestimation for certain storm events (Hurricanes Sandy and Isaias), suggested the need for more nuanced model adjustments based on specific flood scenarios and their characteristics, such as storm tracks and primary driving factors like precipitation and storm surge. The related discussion about when our model performs better has been expanded in Lines 670 to 678:

"The other reason why the model transferability was most successful for Hurricane Irene was that the event mainly driven by significant rainfall, similar to Hurricane Ida (the event that the model was trained for). In contrast, the model performed worse for Hurricanes Sandy and Isaias because these events were mainly driven by storm surge. The original model, considered lower importance

for storm surge, was not effective in predicting the water depths in Sandy and Isaias. In fact, here we see another significant advantage of strategically using physically meaningful features rather than the more commonly used black box approach. By considering the physical phenomena in our model development, we can better understand its strengths and weaknesses and more effectively evaluate its performance."

In Lines 687 to 694:

"The study underscored the complexity of efficiently predicting water depths for major hurricanes and emphasizes the necessity of refining models for better performance during such extreme events. It highlighted the importance of deeper analyses into features causing prediction discrepancies and suggested that addressing different flood types (fluvial vs. storm surge) separately can enhance the model performance. This approach, alongside adjustments for specific flood characteristics like storm tracks and similar influential factors that are distinct for each event, can improve the performance of hindcast models, aiding in the development of more transferable ML-based models."

4. Line 643 – review tense. The sentence does not seem to be incorrect in tense: "This pattern aligns with the southward total slope aspect, where the upper point of the river tends to have shallower depths and the mainstream exhibits deeper water depths."

Response: We have removed the sentence as our model and the analyses have been changed based on the reviewers' comments.

5. Review Legend in Figure 6 (lowest interval overlaps with the next)

**Response:** This figure has been removed and instead we added MDAE based on the earlier comment by the reviewer. We limited the discussion for the $F_Q$ metric in the revised manuscript.

6. It was useful to see the datasets and sources/references summarized in a table (e.g. Table 3).

**Response:** Thank you.

7. Perhaps a single note of abbreviations only at their initial mention is needed – e.g. for machine learning.

**Response:** We have reviewed the manuscript to ensure that all abbreviations, including those for machine learning (ML), are only spelled out at their initial appearance. The exceptions for this are figures, tables and headings that should stand alone.

---

## Author Comment (AC2)

=== General Comments

Disclosure: The reviewer is a ML/AI expert with very limited expertise in the Earth Sciences.

The paper showcases the application of an ANN (likely, a multi-layer perceptron) to hindcast maximum flood depth in a HUC6 watershed in in Northeastern US based on a broad collection of (geographic. hydrologic, meteorologic, topographic, etc.) features, which were determined via a feature selection process. The model was trained on one major flood event and its "transferability" to other major flood events was evaluated yielding, in general, positive results. The paper claims, in essence, the following main contributions: (i) predicting maximum flood depth instead of presence/absence of inundation that has been examined in the literature so far, (ii) appraising the usefulness of a data-driven model on major flood events not used in training/calibration ("transferability").

According to the reviewer's opinion, strengths and weaknesses of this work are as follows:

Strengths:

+ The broad range of features that has been considered in this study.

+ The discussion in subsection 4.2.2 about feature importance (pertains to explainability) seems valuable. Such discussions are often missing from similar works.

**Response:** Thank you.

Weaknesses:

- The paper's motivation is rather weekly framed and its contributions seem rather of limited extent. Why would this work be valuable to Earth scientists?

**Response:** We respectfully disagree with the notion that our contributions are of limited extent. Flood modeling serves as a vital necessity in societies, particularly as changing climate poses increasing threats to lives and properties annually. The complexity of flooding phenomena, driven by various factors, underscores the importance of ongoing research in prediction and forecasting, especially for extreme storm events. While modern ML techniques in earth system modeling, including flooding, have received attention, many adopt them as black box approaches. Our argument advocates for a deeper focus on physical meaning, as evidenced by our successful transferable ML implementation. Transferability to other events, crucial for models to predict events beyond their training data, addresses a significant concern in the field surrounding machine learning models' reported shortcomings for overfitting to training data as noted by proponents of physics-based models. In our approach, we address these concerns by prioritizing the development of models that are transferable across storm events with distinct drivers, thus ensuring their effectiveness in predicting a wide range of events. By working towards bridging the gap between traditional physical modeling and modern machine learning approaches, we aim to enhance the reliability and applicability of flood prediction models.

- Reproducibility of this work seems to be rather low, as important modeling details are not provided. Also, the code and data have not been made publicly accessible.

**Response:** We acknowledge the importance of reproducibility in scientific research and are committed to enhancing this aspect of our work. To address the concerns raised, we made our code with detailed comments for clarity and data publicly accessible at GitHub: (https://github.com/mpakdehi/ANN_MLP-flood-depth-model). The files are usable via open-source software (csv and Jupiter files), ensuring that our work can be fully replicated and scrutinized by the flood modeling community.

 - Lack of a baseline model/approach that can be compared against. How can one tell whether the prediction quality reported in this work is up to par without any comparisons to established approaches? Perhaps, a natural candidate here would have been a physics-based or hybrid (data-driven physics-based) model.

**Response:** We appreciate this constructive feedback. To address your comment, we developed a linear interpolation model as the benchmark model and compared our ML model against its results. Our preliminary results showed that our ML model substantially outperforms the linear interpolation model (Table R1).

The primary focus of this research was not to compare different types of ML models though. Instead, our aim was to test a hypothesis regarding the efficacy of strategically selecting features with specific physical and problem-related significance within a certain type of ML model. We investigated whether such a tailored approach can enhance an ML model performance when applied to different hurricane events and locations, thus assessing its transferability across space and time. This approach contrasts with the common practice of using all available data indiscriminately for training, regardless of their relevance.

A simple, yet practical method for estimating water elevations during flood events is through linear interpolation between observed water depths at nearby gauges upstream and downstream. We evaluated the performance of our ML model by comparing it with this approach along a segment of the Rampao River within our study watershed. This river segment includes five stream gauges in proximity, making linear interpolation suitable.

Our methodology involved systematically removing observations from internal points, one-at-a-time, and then using linear interpolation between the remaining upstream and downstream observed values to estimate the water depths. We then calculated the error compared to the known value at the removed point. Subsequently, we compared the maximum error of this operation for each event with the performance of the ML model, which was trained solely on data from Hurricane Ida but applied to three other events.

Our results demonstrated that the ML model trained for Hurricane Ida consistently outperformed linear interpolation based on the observed values of other events (Table R1). This indicated that the trained ML model can be successfully transferred to different flood events.

Table R1. Maximum errors in hindcasting the observed water depths using our ML model trained for Hurricane Ida and a linear interpolation during other events in a segment of the Rampao River in our study watershed.

| Event | Maximum error of the linear interpolation model | Maximum error of our ML model (trained for Hurricane Ida) |
|---|---|---|
| Hurricane Isaias | 20.6% | 4.9% |
| Hurricane Sandy | 21.2% | 4.3% |
| Hurricane Irene | 17.0% | 2.9% |

We are in the process of preparing a paper related to comparisons of ML models, linear interpolation and hydrodynamic models for hindcasting flood depths during major events. As such, we believe our preliminary analyses are only required in the response document. Language was added to discuss this in the revised paper (Lines 980-986):

"We recommend that future work compares the performance of our ML-based model to traditional physically-based and morphologic-based models using the same datasets. By evaluating the performance, generalizability, and computational efficiency of our ML-based model versus these traditional modeling approaches, we will be able to better validate the strengths of our data-driven methodology. Detailed error analyses between the approaches can also reveal insights into where additional physics knowledge needs to be incorporated into the ML-based model structure and training to improve performance."

- There is a lack of rigor surrounding several ML-related concepts/topics discussed in the paper.

**Response:** We recognize the need for a more rigorous presentation of ML concepts discussed in our manuscript. To rectify this, we enhanced the clarity and depth of our explanation on key ML methodologies employed, ensuring that our discussion adheres closely to established standards and practices within the ML community in SM1.

Based on all this, the reviewer's ratings are as follows:

* Scientific Significance: 3 (Fair)

* Scientific Quality: 2 (Good)

* Presentation Quality: 2 (Good)

Finally, the reviewer hopes that the provided feedback will be useful towards improving this paper.

Response: We thank the reviewer for the constructive comments. Your review has helped us improve the manuscript quality. Detailed responses are provided to your questions. Blue text shows our response and black text shows your comments.

In the revised manuscript, we have added new input features, conducted new analyses to evaluate the feature importance, performed new modeling to improve the performance and expanded our discussion to provide more insights about our results.

=== Specific Comments

1) Lines 101-103: "ML and DL models can make satisfactory predictions (in terms of minimum error in estimating flood characteristics like depth) and generate valuable insights."

By nature, most general purpose DL architectures are notoriously opaque in terms of being interpretable. Hence, it would be very useful to see here some examples of such insights.

Response: This is an insightful comment that is also related to our study contribution. There has been a growing interest in using ML/DL models for modeling flood events. However, there are several issues with these models, including interpretability that was mentioned by the reviewer. A

major issue, which our study investigates, is whether an ML model trained and validated based on a historical event, can be successfully used for prediction of other events (out-of-sample/unseen). This is key because flood models are developed for predicting out-of-sample/unseen events.

To address your comment, we have provided some interpretation techniques (Lines 94-97):

"While there are several issues with these models, including interpretability, techniques such as SHapley Additive exPlanations (SHAP) can enhance understanding of these models' decision-making processes (Lundberg and Lee 2017; Abdollahi and Pradhan 2021). These models enable the identification of key features driving flood characteristics."

And expanded the discussion in the introduction and discussion sections (Lines 106-112).

"Another limitation of these ML studies is the reliance on a single event for training and validation. As such, it is unclear whether a trained and validated model can satisfactorily predict other flood events. These limitations call for studies that evaluate more complex methodologies and a broader range of scenarios on the effectiveness of ML algorithms for predicting flood characteristics. These limitations call for studies that evaluate more complex methodologies and a broader range of scenarios on the effectiveness of ML algorithms for predicting flood characteristics."

Lines 783-790:

"This paper developed an ML-based model for hindcast maximum water depths to address two major limitations of past research in applying ML models for flood estimations: solely predicting flood status (classification-based models) and debate on the transferability of these models across events. We used ANN-MLP to hindcast maximum water depths over an event on a coastal watershed, which is affected by fluvial and tidal floods. The model was informed by underlying

physical flood processes and initial conditions (in the watershed and rivers), represented through a set of features (geographic location, topographic, climatic, land surface, hydrologic, hydrodynamic and soil)."

2) Lines 114-115: "... and different ML models were examined on the same dataset. "

Examined in what sense?

**Response:** Language was added to clarify the statement (Lines 102-106). The examination of different ML models in previous flood modeling studies refers to the evaluation of their predictive performance. This involved comparing the predictive accuracy of various algorithms by training them on a portion of the observed dataset (training set) and then testing their predictions on the remainder of the dataset (validation or test set). This approach helps in identifying the most effective ML models for flood prediction based on performance metrics, such as accuracy and precision. The revised text is provided here for your convenience:

"The datasets were often split into two subsets, and ML models were examined trained on a portion of the dataset (training set) and then tested for the remainder of the dataset (validation or test set). This approach helps in identifying the most effective models for flood predictions based on performance metrics, such as recall or the area under the Receiver Operating Characteristic (ROC) curve."

3) Lines 182-184: "Next, we assessed the transferability of our developed model across three other extreme events — Hurricanes Isaias, Sandy, and Irene — in the same watershed."

Is "transferability" the right term to use here, instead of "generalization ability" or something equivalent? In other words, is "transferability" used in place of out-of-sample (test) performance?

**Response:** Physical earth system modeling specially for extreme events are notorious for lack of transferability. This is in fact one of the points that make contributions of our paper valuable. The term "transferability" in this paper is used in place of out-of-sample performance (unseen data). While "generalization ability" speaks to a model performance on out-of-sample data that is, in theory, drawn from the same distribution as the training data, "transferability" goes a step further by evaluating the model performance across different flood events (unseen data), potentially with their unique characteristics (Jiang et al. 2024; Wagenaar et al. 2018). The "transferability" represents the model applicability across different flood events, not just in handling unseen data from the same event that it was trained and tested for. This is consistent with the goal of model development in all flood prediction studies; i.e., the models are validated to be applied for future events that are unseen in the historical observations. The language is added to the manuscript (Lines 147-150):

"The term "transferability" refers to the model's ability to predict different flood events beyond the scope of its training data, validating its applicability to unseen scenarios, potentially with their unique characteristics (Jiang et al. 2024; Wagenaar et al. 2018)."

4) Lines 386-387: "The observed flood data and features were split into training and testing sets, with 70% to 90% of the data used for training and 10% to 30% for testing (Joseph 2022; Nguyen et al. 2021)."

How many samples were used for training and how many for testing and why such a split was selected? Also, Line 403 states "We allocated 90% of the data for training and 10% for testing." which conflicts with what is said here.

**Response:** Our model evaluation included multiple efforts. We examined multiple split scenarios for the flood events (observed at 116 stream gauges), ranging from 70-30 to 90-10 (Lines 379-387):

"The observed water depths and features were split into training and testing sets, with 70% to 90% of the data used for training and 10% to 30% for testing as suggested by Joseph (2022) and Nguyen et al. (2021). After exploring various splits within the 70% to 90% range for training data, the 90% allocation for training (104 out of 116 stream gauges) was determined to be optimal for our specific dataset and model based on preliminary testing, the model complexity, and the desire to maximize the amount of data used for training while still retaining satisfactory results for the test phase (12 out of 116 stream gauges). While the train percent (90%) seems high and suggests potential for model overfitting, this same model was most successful in the transferability across other three flood events (out-of-sample)."

5) Lines 391-394: "We used the Random Search cross-validation approach (Boulouard et al. 2022; Hashmi 2020) to perform hyper-parameter optimization. This approach performs a randomized search on hyperparameters using cross-validation. The hyperparameters we optimized here included the number of layers, units, activation functions, optimizer, regularization rate, batch size, and epochs. "

As there are several hyper-parameters involved, a bayesian search method would be more pertinent and fruitful than a plain random search. There are several handy Python-based implementations out there that could have been employed (e.g., see https://optuna.org/).

**Response:** Based on the reviewer suggestion, we explored Bayesian search method and found both methods (random search and Bayesian search) converged on comparable optimal hyperparameters, affirming our model robustness. Based on the reviewer comments, we now only present the Bayesian search results in the revised paper. In Lines 408-413:

"Bayesian search offered a targeted search based on probabilistic modeling, iteratively refining the search area based on past evaluations to efficiently select the most promising hyperparameter sets. The selection of the optimal hyperparameters was guided by minimizing the cross-validation MSE, ensuring the chosen configuration significantly improved the model predictive performance for maximum water depths."

We have also added the following in Lines 574 to 576 of the revised manuscript:

"In the development of our ANN-MLP model for hindcasting maximum water depths during Hurricane Ida, we used Bayesian search with a cross-validation strategy for hyperparameter optimization. Details of the optimization can be found in SM1."

6) Lines 400-401: "Cross-validation was performed using a 5-fold cross-validation strategy during the hyperparameter optimization process."

Why was 5-fold cross-validation used in favor of hold-out validation, which uses a separate, dedicated validation set?

**Response:** Explanation was added to provide the reasons for this choice in the revised manuscript (Lines 400-407):

"Cross-validation, particularly through methodologies like the Prediction Sum of Squares criterion for predictor selection and for parameter estimation and predictive error assessment, has been foundational in improving predictive models. This approach distinguishes between model selection and assessment (Allen 1974; Geisser 1975; Stone 1974). Cross-validation was performed using a 5-fold cross-validation strategy during the hyperparameter optimization process. Opting for 5-fold cross-validation over hold-out validation in our hyperparameter optimization process reflects a balance between comprehensive model evaluation and computational efficiency."

7) Lines 534-536: "Rain-MAX" and "Rain-Mean" suggested that they offer similar information about maximum and average rainfall values across the watershed. Consequently, "Rain-Mean" was excluded from consideration."

Shouldn't "Rain-Mean" to be retained instead as being potentially much less noisy than "Rain-MAX"?

**Response:** In the revised manuscript, new modeling has been conducted and rain data was modified, by considering all feature selection methods and adding forward feature selection including interactions among the input features. We have excluded both 'Rain-MAX' and 'Rain-Mean' from our model and kept "Rain_Point" that is local rainfall depths at stream gauges by our new feature selection method that considers interactions among the features. The related revisions are provided here (Lines 542-570):

". Results and discussion

4.1. Feature selection

Using Pearson's correlation analyses, we eliminated five features with absolute correlation coefficients >0.70, the cutoff threshold suggested in previous studies (Cao et al. 2020; Chen et al. 2023; Lee et al. 2020). According to Figure 5, the strong correlation coefficient of 0.99 between drainage area and flow accumulation, indicated that both features capture similar information about water flow and storage in the watershed. To avoid collinearity issues, flow accumulation was excluded from further analyses due to its weaker correlation with flood depth. Similarly, features that demonstrated weaker correlations with flood depth or were highly correlated with multiple features, were excluded. These analyses ensured that independent variables, which are essential for modeling maximum water depths, are retained in our modeling.

Figure 5. Heatmap of Pearson correlation matrix for the initial model features.

Next, we conducted PCA to assess the importance of the features retained by Pearson's correlation analyses in hindcasting maximum water depths. The analyses showed that the slope at the stream gauge, slope aspect, slope invariability, curvature at the stream gauge, and average curvature across the contributing watershed were the least important features for capturing the overall variability of maximum flood depth. Consequently, we excluded these features from our analyses. The lesser importance of slope at the stream gauge and slope aspect may be since river slope is related to bathymetry, which is typically not represented well by DEMs (Bhuyian and Kalyanapu 2020).

The forward feature selection method showed that initial water depth, elevation, TWI, antecedent soil moisture, rainfall, and distance from storm surge at the stream gauge (all point-based), as well as average storm surge and maximum wind speed across the contributing watershed, along with their interactions were selected for the final ML model. Considering the interactions among the features improved the model performance. This was expected because a combination of some of the features better explain the underlying physical processes. For instance, using the combination of storm surge and TWI as one unified feature can be an indication of the physical propagation of storm surge that occur primarily in waterways."

8) Lines 554-555: "The optimization process involved 500 fits, with each fit considering 100 candidates for each of the five folds in the cross-validation."

It would be helpful if much more explanation were provided here.

**Response:** We have replaced the random search optimization method with Bayesian search optimization based on your earlier comment. We have elaborated on this in the revised manuscript (Lines 632 to 659):

"In the development of our ANN-MLP model for hindcasting maximum water depths during Hurricane Ida, we used Bayesian search with a cross-validation strategy for hyperparameter optimization.

We defined a broad search space (pbounds) encompassing the number of layers, setting a range between 1 and 3 layers with the units varied from 10 to 90, regularization rates (0.01 to 0.2), optimizers (Stochastic Gradient Descent (SGD) (Bottou 2012) and Adaptive Moment Estimation

(Adam) (Singarimbun, Nababan, and Sitompul 2019)), and activation functions (Exponential Linear Unit (ELU) (Trottier, Giguere, and Chaib-draa 2017), and Rectified Linear Unit (ReLU) (Agarap 2019)) in hidden layers, facilitating a thorough exploration of model architectures. A linear activation function was used for the output layer. The batch size, determining the number of samples processed before the model updates its parameters, varied between 4 and 16, providing a balance between training speed and memory usage. Lastly, the number of epochs, which dictates the number of complete passes through the training dataset, was explored from 100 to 1000, to find the optimal duration for model training. The optimization process, implemented via the Bayesian search framework, systematically evaluated combinations of hyperparameters across the defined space. It began with 2 initial random evaluations (init_points=2) of hyperparameter sets, followed by 3 guided evaluations (n_iter=3). Thus, a total of 5 unique hyperparameter sets were assessed. Utilizing cross-validation, the dataset was divided into three subsets or 'folds. For each iteration of the optimization process, a different fold was held out as the validation set, while the remaining folds were used for training the model. For each set, we applied 5-fold cross-validation (cv=5), resulting in each set being evaluated 5 separate times, one for each fold. Consequently, there were 5×5=25 individual model trainings during the optimization process. This approach ensures that each data point contributes to both the training and validation phases, enhancing the reliability of the performance assessment. The Bayesian search process with a cross-validation strategy culminated in identifying an optimal set of hyperparameters that significantly enhanced the model predictive performance. The optimized configuration comprised a specific arrangement of number of layers, units, epochs, batch size, a precise regularization rate, and an optimal combination of optimizer and activation function, tailored to maximize the accuracy of estimations of maximum flood depth."

9) Lines 560-562: "The best hyperparameters were identified as follows: 50 units, a regularization rate of approximately 0.104, the sgd optimizer, one layer, 600 epochs, a batch size of 8, and the elu                                                  activation                                                  function."

What was the precise hyper-parameter search domain (e.g., range of number of units, choices of optimizers, etc.)? Also, acronyms such as "sgd" and "elu" should be spelled out somewhere. Regarding the latter, is this the activation function used in the hidden layer? How about the output layer? Does it use a linear activation function? Finally, the conclusion is that a one (hidden) layer network is the best model as estimated via cross-validation; this is somewhat surprising and, curiously enough, is not commented on.

**Response:** We have replaced the random search optimization method with Bayesian search optimization based on your earlier comment. It was not surprising that cross-validation indicated a one-layer model as the best option, given our limited number of samples (116). We have elaborated on this in the revised manuscript (SM1).

10) Section 4.3. "Examining the machine learning (ML) model transferability across flood events"

What motivates this "transferability" study? Only intellectual curiosity? Why not train with data from all three identified flood events? Why wouldn't (or would, for that matter) someone expect that the model, trained on a single flood event, will also perform well for other floods in the same watershed? Are there important variables not captured by the features considered in this study? And, if so, why were these omitted?

**Response:** There has been a growing interest in using ML/DL models for modeling flood events. However, there are limitations with these models, including transferability across events, which our study focuses on. The process of predicting flood characteristics in a given watershed via our model is that the model is first trained and validated based on one event and then this trained and validated model will be applied to predict other flood events without retraining. Our question is whether an ML model trained and validated based on a historical event, can be successfully used for prediction of other events (out-of-sample/unseen)? This is key because eventually flood models are developed for predicting out-of-sample/unseen events. Thus, training a model, regardless of the number of events (one or three), is not the issue. The issue is whether ML models can be successfully applied to other models. This is the research that we are experimenting when discussing model transferability. It is considering suitability, feasibility, efficiency, and parsimony of ML models that the modeling community has been investigating without investigating their suitability beyond what they are trained for. The detailed explanation has been added to the revised manuscript to clarify the importance of the model transferability (Lines 142-161):

"Despite previous efforts, the development of computationally efficient and user-friendly flood prediction models remains a challenge. ML-based models, although promising and computationally efficient, have not gained widespread acceptance among practitioners due to concerns about their reliance on predicting flood characteristics for other events (out-of-sample). Transferability is particularly crucial given the growing reliance on ML modeling methods, like ANNs, as suggested by Wenger and Olden (2012). The term "transferability" refers to the model's ability to predict different flood events beyond the scope of its training data, validating its applicability to unseen scenarios, potentially with their unique characteristics (Jiang et al. 2024; Wagenaar et al. 2018). Furthermore, there has yet to be research investigating the extent to which

flood depths prediction models can be transferred and applied successfully to different events beyond the initial training settings. It, therefore, remains unclear whether an ML-based model, which is trained, validated, and tested against a historical event, performs satisfactorily in predicting flood characteristics of other events in the same watershed. Floods originate from various sources and the flood characteristics depend on the unique characteristics of storm events. High wind events tend to generate storm surges that move upstream, while intense rainfall over upstream watersheds leads to fluvial flooding that moves downstream towards the coast. Conversely, slow-moving storm systems can cause intense local rainfall, resulting in overland runoff entering rivers along their paths rather than a concentrated upstream inflow flood wave. Hence, it is crucial to avoid overfitting an ML model to a single historical flood event, as it can lead to significant underperformance in handling other events."

=== Technical Corrections

1) Overall: The paper constantly mentions that it employs an "ANN," which is a non-specific term. I suspect that it uses a multi-layer perceptron and, if so, the paper needs to reflect this.

**Response:** 'ANN' has been replaced by 'ANN-MLP', where applicable, to specify the type of neural network employed. These changes appear throughout the manuscript.

2) Line 89: "Machine learning (ML) and deep learning (DL) models..."

As DL models are ML models, perhaps, this could be slightly rephrased like "Machine learning (ML) and, in particular, deep learning (DL) models, ..."

**Response:** The text has been revised to the following to address your comment (Lines 82-83):

"ML and, in particular, deep learning (DL) models, offer an alternative approach that can rapidly capture complex relationships between various influencing factors and flood characteristics."

3) Lines 95-96: "..., and through their intricate nonlinear structures and algorithms."

Consider rephrasing this.

**Response:** The description in Lines 86-88 has been revised to better articulate the complex architectures and computational strategies of the models in question.

"These models mathematically represent the nonlinearity of flood dynamics with pertinent features and observed flood data using complex nonlinear structures and algorithms."

4) Line 105: "...neural networks (ANNs), random forest, convolutional neural networks..."

Please note that convolutional neural networks are ANNs.

**Response:** The term "CNN" has been removed from the revised manuscript.

5) Lines 180-181: "The developed ML-based model combined the ANN algorithm with feature selection methods and geospatial data."

Perhaps, this should be rephrased as "The developed ML-based model combined an ANN with feature selection methods and geospatial data."

**Response:** The phrasing in Lines 175-177 has been adjusted following your suggestion:

"Our study developed a modeling framework based on an ML algorithm, Multi-Layer Perceptron (MLP) architecture for our ANN model. This algorithm was coupled with feature selection methods and geospatial data."

6) Line 299: "... remove any fake depressions, ..."

Perhaps, rephrase to "... remove any spurious depressions, ..." or something similar.

**Response:** To enhance the technical accuracy of the manuscript, the term 'fake depressions' has been replaced with 'spurious depressions' (Lines 288-291):

"To remove any spurious depressions, the DEM sinks were filled to account for artificial depressions that can impede the realistic simulation of water flow, ensuring that the derived water pathways and other hydrologic computations reflect true surface conditions (Khosravi et al. 2018; Zhu et al. 2013)."

7) Lines 350-351: "The PCA components were evaluated based on their absolute values, allowing us to quantify the contribution of each feature to the overall variance."

This statement is somewhat confusing and could be improved. PCA components are the eigen-vectors of the data's covariance matrix. What does it mean to take their absolute value and how does this help in quantifying the contribution of each feature to the overall variance? The latter is typically accomplished by considering he corresponding eigen-values, which are necessarily non-negative.

**Response:** To elucidate the approach taken in our PCA analyses, the manuscript has been revised to clarify that we examined the eigenvalues of each principal component to quantify their contribution to the dataset's overall variance (Lines 339-341):

"Next, PCA was applied to the features retained after the Pearson's correlation analysis. In the PCA method, the contribution of each feature to the overall variance is quantified by examining the eigenvalues associated with each principal component."

8) Line 354: "... the PCA captures both linear and non-linear relationships."

PCA is considered a "linear (affine)" method, as it assumes that the features are a linear-affine function of some latent variables. What is meant by non-linear relationships here?

**Response:** The manuscript has been revised to clarify that while PCA is inherently a linear method, it is adept at finding patterns in the data. The previous mention of 'non-linear relationships' has been revised to avoid any misinterpretation of PCA's capabilities (Lines 341-345).

"Compared to the Pearson's linear correlation, the PCA can reveal underlying patterns or structures in the data that are not immediately apparent. PCA allows us to understand how much variance each principal component considers in the dataset, providing a clear measure of feature significance in terms of explaining the data variance."

9) Lines 356-357: "Through PCA, we determined which principal components in the feature set captured the most variation."

This is the very definition of what constitutes a principal component, so it is redundant to state.

**Response:** Upon review, we have more efficiently discussed the PCA function in identifying principal components that capture the most variation, streamlining the content for better readability and precision. The revised statement is provided here for your convenience (Lines 330-356):

"We employed multiple feature selection methods Pearson's correlation coefficients (Cao et al., 2020; Chen et al., 2023; Lee et al., 2020) and PCA—a widely used technique in many ML modeling studies (Abdrabo et al., 2023; Chang et al., 2022; Reckien, 2018)—and forward feature selection that accounts for interactions among the model features. We applied a step-by-step approach to utilize these three techniques.

First, the Pearson's correlation coefficients were used to assessing the linear relationships among the features and target variable. The strength and direction of linear relationships were evaluated using Pearson's correlation coefficients. These analyses enabled us to narrow down the initial list of the features.

Next, PCA was applied to the features retained after the Pearson's correlation analysis. In the PCA method, the contribution of each feature to the overall variance is quantified by examining the eigenvalues associated with each principal component (Abdrabo et al. 2023). Compared to the Pearson's linear correlation, the PCA can reveal underlying patterns or structures in the data that are not immediately apparent. PCA allows us to understand how much variance each principal component considers in the dataset, providing a clear measure of feature significance in terms of explaining the data variance. By aggregating the absolute values across all features, we obtained the importance for each feature, which enabled us to rank them in a descending order and omit least important features.

Last, the forward selection method was applied on the features retained. This method then incrementally added variables, weighing both their individual impact and interactions, enhancing the model predictive performance by focusing on features with substantial influence on flood depths (Macedo et al. 2019; Horel and Giesecke 2019; Macedo et al. 2019). This method adds variables to a model based on their predictive power. This iterative process starts with no variables and includes the most predictive one at each step, considering both its individual impact and its interactions with already included variables. This selection continues until adding more features does not significantly enhance the model performance metric in terms of Akaike Information Criterion."

10) Lines 368-370: "One of the key advantages of using ANN is its capacity for generalization, as highlighted by Maier et al. (2023), allowing the model to perform well on unseen data, making it robust and reliable for real-world flood estimations."

As any reasonably selected/constructed model is capable of generalization (provided it is well-parameterized), this statement here should probably be removed or rephrased.

**Response:** The manuscript has been revised to reflect a better understanding of the model generalization (Lines 365-370):

"One of the strengths of using ANNs in modeling tasks like flood predictions is their notable flexibility and capability to approximate complex, non-linear relationships, potentially enhancing their performance for unseen data. It is essential, however, to acknowledge that the capacity to generalize depends on selecting relevant features that explain the underlying physical processes and the spatiotemporal variability, model selection, parameterization, and training the model."

11) Lines 373-379: "ANNs are computing systems inspired ... based on the input data it is processing (McCulloch and Pitts, 1943)."

To conserve space, this paragraph could be pruned, as it talks about very well-known facts about neural networks.

**Response:** Following your comment, we have removed the paragraph in the revised manuscript.

12) Lines 387-389: "The numerical features in the data were standardized using the StandardScaler function from the Scikit-learn library of python."

Instead of referring to a software implementation here, it would be more helpful to describe the type of scaling in a couple of sentences.

**Response:** To improve the clarity, we have rephrased the text about feature standardization. The manuscript now explains that this process normalizes the features, which is essential for effective training of gradient descent-based models (Lines 393-397):

"In preparing our dataset for the neural network model, numerical features were standardized to have a mean value of zero and a standard deviation of one. This scaling process ensured that each feature contributes proportionately to the model predictions, mitigating the potential bias towards variables with larger scales."

13) Lines 389-391: "Hyperparameter optimization is a step in improving the performance of ML models. This process involves identifying the optimal hyper-parameter values for ML classifiers."

Strictly speaking, it is synonymous to model selection based on generalization performance. As a matter of fact, the paper points this out shortly after. Also, hyper-parameter optimization can be applied to any model family regardless of what task it addresses, not only to ML-based (i.e., data-driven) classifiers.

**Response:** The text has been revised to communicate that hyperparameter optimization is an integral part of the model development across all types of models, not limited to ML classifiers (Line 398-399):

"Hyperparameter optimization is a step in improving the performance of ML models. This process involves identifying the optimal hyper-parameter values."

14) Lines 391-394: "rs. We used the Random Search cross-validation approach (Boulouard et al. 2022; Hashmi 2020) to perform hyper-parameter optimization. This approach performs a randomized search on hyperparameters using cross-validation. The hyperparameters we optimized here included the number of layers, units, activation functions, optimizer, regularization rate, batch size, and epochs. "

In the first sentence, only Hashmi 2020 seems to be useful, unless the sentence is framed differently (e.g., "as also adopted by Boulouard et al. 2022"). The second sentence seems redundant.

**Response:** We have omitted this part as we decided to use Bayesian search method based on your earlier suggestion.

15) Line 395: "The best hyperparameters were selected based on the negative mean squared error."

While this may be true, it may be better to plainly state that "hyperparameters were selected based on cross-validation MSE."

**Response:** The criterion for selecting hyperparameters has been clarified as 'cross-validation mean squared error (MSE)', which more accurately reflects the methodological rigor of our optimization process (Lines 410-413).

"The selection of the optimal hyperparameters was guided by minimizing the cross-validation MSE, ensuring the chosen configuration significantly improved the model predictive performance for maximum water depths."

16) Lines 406-407: "This allocation of 10% for testing, combined with these methodologies, is designed to enhance the model's ability to generalize across diverse scenarios."

How is this statement substantiated? A test set's purpose is to honestly appraise a model's generalization performance. It is not supposed to be leveraged in any way to train/select models according to their generalization abilities.

**Response:** The purpose of allocating 10% of data for test phase has been clarified to reflect its role in providing a fair assessment of the model performance on unseen data (transferability), not influencing the model training or selection process (Lines 387-393):

"The allocation of 10% of the data for testing serves to provide an unbiased appraisal of the model generalization performance after training and hyperparameter optimization. This evaluation process, complemented by methodologies such as cross-validation and hyperparameter optimization, is structured to identify a model configuration that is likely to perform well across

unseen data. This approach aims to ensure that the final model, selected based on its performance on the validation set during hyperparameter optimization, is tested on entirely unseen data to confirm its generalization ability."

17)    Line    423:    "2.2.4.    Model    interpretation    "

Perhaps, in order to preserve the nuanced distinction between model explainability versus interpretability, "Model Explainability" is more pertinent as a heading, given the content of this subsection. Same comment applies to line 574.

**Response:** We have revised the subsections to 'Model explainability' following your suggestion (Lines 441 and 584):

"2.2.4. Model explainability"

"4.2.2. Model explainability"

18)    Line    563:    "This    meticulous    hyperparameter    optimization    approach..."

I am unsure if a (plain) random hyper-parameter search can be regarded as "meticulous," when, in theory, one could do an exhaustive grid search.

**Response:** We have removed the Random Search approach and used Bayesian search according your earlier comment.

19) Lines 576-577: "The SHAP values measure the contribution of a feature to the estimation for each sample in comparison to the estimation made by a model trained without that feature."

Maybe "estimation" needs to be replaced with "prediction" here.

**Response:** The statement has been revised by replacing "estimation" to "prediction" (Lines 447-449):

"In other words, SHAP estimates how much each feature contributes to the model prediction output for a particular instance."

20) Figure 5: Some legends feature an underscore, which should be removed.

**Response:** All underscores in the Figure 7 (formerly Figure 5 in the original manuscript) have been removed following your comment (Line 604). The revised figure is provided here for your convenience:

[Figure]

Figure 7. Aggregated Shapely additive explanations (SHAP) feature importance radar plot of the maximum water depth model.

21) Line 581: "The most influential features in estimating flood depths are antecedent water level..."

From where is this concluded? I do not see "antecedent water level" as a label in Figure 5.

**Response:** We have ensured that the terminology used in Figure 7 (formerly Figure 5 in the original manuscript) matches the text description by replacing 'Mean_GaugeHeight' with 'Initial water depth' for clarity and consistency (Line 736). The revised figure is provided here for your convenience:

[Figure]

Figure 7. Aggregated Shapely additive explanations (SHAP) feature importance radar plot of the ML model for hindcasting maximum water depths.

22) Line 679: "...hyperparameter set was used as the optimal parameterization scenario."

What is meant by this?

**Response:** The sentence has been revised for clarity (SM1):

"The optimal hyperparameters identified through the Bayesian optimization method included one hidden layer with 47 units, 636 epochs, a batch size of 8, a regularization rate of approximately 0.07, the SGD optimizer, and the ELU activation function. However, after manually adjusting the number of units to 50 and the regularization rate to 0.104, we achieved the best performance.

Additionally, we implemented early stopping, a technique designed to halt the training process when model performance no longer improves on the training and test datasets, further enhancing our ANN-MLP model."

23) Lines 679-681: "This deterministic approach does not incorporate the uncertainty from model parameterization. Probabilistic models are needed to address this uncertainty."

What is meant by "uncertainty from model parameterization"? Is it "uncertainty from model misspecification"? Also, if that is so, how can probabilistic models address this? Some details are warranted here.

**Response:** The term "uncertainty from model parameterization" refers to the variability in model predictions that arises due to the choice of the model hyperparameters derived during the model tuning. This is distinct from "model misspecification" that pertains to errors or inaccuracies in the model structure itself, such as incorrect assumptions about the relationship between variables or the distribution of errors. The revised manuscript now includes additional details to clarify 'uncertainty from model parameterization', distinguishing this concept from model misspecification and explaining how probabilistic models can tackle these uncertainties (Lines 709-722):

"Parameterization uncertainty acknowledges that the exact values of model parameters (e.g., weights in an ANN-MLP) determined through training may not perfectly capture the true underlying processes, leading to variability in our predictions. Probabilistic models address this uncertainty by incorporating it directly into the modeling process, offering a range of possible outcomes with associated probabilities (posterior probability distributions) rather than a single

deterministic output. This is achieved through techniques like Bayesian inference, where prior knowledge about parameters is updated with observed data to produce a posterior distribution of parameters. This approach provides a more nuanced understanding of uncertainty, allowing predictions to reflect both the variability observed in the data and the confidence in the model's parameter estimates. To address the limitations of deterministic models, like the ANN-MLP used in this study, future research should explore integrating probabilistic modeling techniques such as Bayesian inference. Exploring alternative data sources and methodologies, such as incorporating spatially detailed features or dynamic time series data, could also help in capturing the complexities of watershed characteristics more accurately."

24) Lines 731-732: "We recommend that future work compares the performance of our ML-based model to traditional physically-based and morphologic-based models using the same datasets."

Why is this recommended and not performed in this study? Besides, the paper reports results against no baseline performance, which is problematic when assessing its usefulness.

**Response:** Your point is an excellent direction for future research, but this comparison was not within the scope of our paper that focuses on evaluating the transferability of ML models in hindcasting maximum river flood depth. As discussed in the introduction section (Lines 185-214), there are broadly three groups of flood models and we focus on evaluating one model group: ML models.

The usefulness of our ML model was assessed against observed flood data at 116 stream gauges in four historical flood events. Our assessments were done quantitatively using various fit metrics

($R^2$, MAE, $F_Q$ and MADE). Yet, the absence of a baseline model in the paper has been acknowledged as a limitation (Lines 958-964):

"We recommend that future work compares the performance of our ML-based model to traditional physically-based and morphologic-based models using the same datasets. By evaluating the performance, generalizability, and computational efficiency of our ML-based model versus these traditional modeling approaches, we will be able to better validate the strengths of our data-driven methodology. Detailed error analyses between the approaches can also reveal insights into where additional physics knowledge needs to be incorporated into the ML-based model structure and training to improve performance."

We are currently conducting comparison among our ML model and a simple linear interpolation model for another paper. Preliminary results for Lower Hudson River Watershed are provided here but we believe this should not be included in the main manuscript as it dilutes the main focus.

The primary focus of this research was not to compare different types of ML models though. Instead, our aim was to test a hypothesis regarding the efficacy of strategically selecting features with specific physical and problem-related significance within a certain type of ML model. We investigated whether such a tailored approach can enhance an ML model performance when applied to different hurricane events and locations, thus assessing its transferability across space and time. This approach contrasts with the common practice of using all available data indiscriminately for training, regardless of their relevance.

A simple, yet practical method for estimating water elevations during flood events is through linear interpolation between observed water depths at nearby gauges upstream and downstream. We

evaluated the performance of our ML model by comparing it with this approach along a segment of the Rampao River within our study watershed. This river segment includes five stream gauges in proximity, making linear interpolation suitable.

Our methodology involved systematically removing observations from internal points, one-at-a-time, and then using linear interpolation between the remaining upstream and downstream observed values to estimate the water depths. We then calculated the error compared to the known value at the removed point. Subsequently, we compared the maximum error of this operation for each event with the performance of the ML model, which was trained solely on data from Hurricane Ida but applied to three other events.

Our results demonstrated that the ML model trained for Hurricane Ida consistently outperformed linear interpolation based on the observed values of other events (Table R1). This indicated that the trained ML model can be successfully transferred to different flood events.

Table R2. Maximum errors in hindcasting the observed water depths using our ML model trained for Hurricane Ida and a linear interpolation during other events in a segment of the Rampao River in our study watershed.

| Event | Maximum error of the linear interpolation model | Maximum error of our ML model (trained for Hurricane Ida) |
|---|---|---|
| Hurricane Isaias | 20.6% | 4.9% |
| Hurricane Sandy | 21.2% | 4.3% |
| Hurricane Irene | 17.0% | 2.9% |